



# Attributing trend in naturalized streamflow to temporally explicit vegetation change and climate variation in the Yellow River Basin of China

Zhihui Wang[1,3], Qiuhong Tang[2], Daoxi Wang[1,3], Peiqing Xiao[1], Runliang Xia[4], Pengcheng Sun[1], Feng Feng[5]

[1]Key Laboratory of Soil and Water Conservation on the Loess Plateau of Ministry of Water Resources, Yellow River Institute of Hydraulic Research, Yellow River Conservancy Commission, Zhengzhou, 450003, China
[2]Key Laboratory of Water Cycle and Related Land Surface Processes, Institute of Geographic Sciences and Natural Resources Research, Chinese Academy of Sciences, Beijing, 100101, China
[3]Henan Key Laboratory of Ecological Environment Protection and Restoration of the Yellow River Basin, Yellow River Institute of Hydraulic Research, Zhengzhou, 45003, China
[4]Henan Engineering Research Center of Smart Water Conservancy, Yellow River Institute of Hydraulic Research, Zhengzhou, 45003, China
[5]Yellow River Conservancy Technical Institute, Kaifeng, 475004, China

*Correspondence to*: Qiuhong Tang (tangqh@igsnrr.ac.cn)

**Abstract.** The naturalized streamflow, i.e., streamflow without water management effects, in the Yellow River Basin (YRB) has been significantly decreased at a rate of $-3.71 \times 10^8$ $m^3 \cdot yr^{-1}$ during 1982-2018 although annual precipitation experienced insignificantly positive trend. Explicit detection and attribution of naturalized streamflow is critical to manage limited water resources for sustainable development of ecosystem and socio-economical system. The effects from temporally explicit changes of climate variables and underlying surfaces on the streamflow trend were assessed using Variable Infiltration Capacity (VIC) model prescribed with continuously dynamic leaf area index (LAI) and land cover. The results show a sharp increase of LAI trend and land use change as a conversion of cropland into forest-grass in the basin. The decrease in naturalized streamflow can be primarily attributed to the vegetation changes including interannual LAI increase and intra-annual LAI temporal pattern change, which accounts for the streamflow reduction of $1.99 \times 10^8$ $yr^{-1}$ and $0.45 \times 10^8$ $m^3 \cdot yr^{-1}$, respectively. The impacts of LAI change are largest at the sub-region of Longmen-Huayuankou where LAI increasing trend is high and land use change is substantial. Attribution based on simulations with multi-year average LAI changes obviously underestimates the impacts of interannual LAI change and intra-annual LAI temporal change on the natural streamflow trend. Overall, the effect climate variation on streamflow is slight because positive effect from precipitation and wind speed changes was offset by the negative effect from increasing temperature. Although climate variation is decisive for streamflow change, this study suggests that change in underlying surface has imposed a substantial trend on naturalized streamflow. This study improves the understanding of the spatiotemporal patterns and the underlying mechanisms of natural streamflow reduction across YRB between 1982 and 2018.



## 1 Introduction

The Yellow River is the second-longest river in China and its contribution to Chinese civilisation has earned it the title of the country's "Mother River". It originates in the Tibetan Plateau, flows through the Loess Plateau and North China Plain, and discharges into the Bohai Gulf, and it has a total length of about 5,464 km and drains a watershed of 752,443 km$^2$ (Tang et al., 2013). It supports 30% of China's population and 13% of China's total cultivated area with water resources accounting for only about 2.6% of China's water (Cuo et al., 2013). Because of less precipitation, there is a critical water shortage

problem in the Yellow River Basin (YRB). The basin has only 620 m$^3$ in per capita water resources, which is 30% and 7.5% of the national and global per capita water resources, respectively (Fu et al., 2004; Bao et al., 2019).

Like elsewhere throughout the world, climate change is taking place in the YRB as reported by previous studies (Fu et al., 2004; Xu et al., 2007; Hu et al., 2011). These studies consistently reported temperature increases, spatio-temporal variations in precipitation in the YRB. Meanwhile, to mitigate the severe soil erosion and deteriorating ecological environment, a series

of soil and water conservation measures and ecological restoration projects have been implemented by Chinese government, including afforestation, Grain for Green Project (GFGP), grazing prohibition, terraces, and check dams (Yao et al., 2011; Jia et al., 2014). In recent three decades, the YRB has experienced drastic change of underlying surface conditions, including land use/cover, vegetation structure, topography and frozen soil, which has significantly altered the evapotranspiration and terrestrial water storage associated with runoff and its routing processes (Cheng and Jin, 2013; Sun et al., 2015; Bai et al,

2019; Yang et al., 2020; Zhai et al., 2021; Wang et al., 2022). A number of observational studies have shown that streamflow in different parts (e.g., source region, Loess Plateau, etc.) of the YRB decreased over the past decades (Tang et al., 2008; Hu et al., 2011; Zhao et al., 2015; Feng et al., 2016; Wu et al., 2018). This may lead to more serious water use conflict between ecosystem and socio-economical system. With the increasing scarcity of water resources, ecologists, hydrologists and decision makers have paid considerable attention to how much of the observed change in annual streamflow of YRB can be

attributed to climate variability and human activities for adaption in future water resources management (Chang et al., 2016; Wu et al., 2018).

Numerous studies have been conducted to investigate the change in river streamflow induced by climate change and human activities under global change (Tang, 2020). Statistical method including double mass curve (Gao et al., 2011) and climate elasticity model (Roderick et al., 2011) was the easy-to-use way to identify the contributions of climate and human impacts

on runoff, whereas it lacks the physical mechanism description and only can assess the overall impact induced by human activities. As the first analytical expression of Budyko's hypothesis was proposed by Fu (1981) according to the hydrological and climatic physical mechanism of the basin, Budyko-based elasticity method have been extensively used in the YRB to quantify the influence of changes in precipitation, potential evapotranspiration and watershed natural features on streamflow (Zhang et al., 2008; Zhao et al., 2014). To further isolate the vegetation effect on the streamflow, the relationship between

watershed feature parameter and vegetation change at catchment scale has been detailed discussed in different basins and





regions in the YRB (Zhang et al., 2016; Bao et al., 2019, Wang et al., 2021). However, above methodology is only able to attribute the multi-year average streamflow change between different periods.

Recently, process-based hydrological models have been used more and more widely due to interannual change of climate variables, vegetation, irrigation, dams and coal mining, etc. can be considered in the model to some extent for quantifying the impacts of various factors on the hydrological process (Tang et al., 2008; Tang et al., 2013; Wang et al., 2017; Luan et al., 2020). Very few studies focused on the impact of intra-annual temporal pattern change of climate variables and vegetation on the streamflow (Tang et al., 2008). Among commonly used models, the variable infiltration capacity (VIC) model is a physically-based macroscale hydrological model developed to solve water and energy balances (Liang et al., 1994, 1996). It has been successfully applied to simulate and attribute natural hydrological processes at both regional and global scales (Matheussen et al., 2000; Haddeland et al., 2006; Xie et al., 2007; Wang et al., 2012; Zhang et al., 2014; Yuan et al., 2016; Zhai et al., 2018; Yao et al., 2019; Zhu et al., 2021). The VIC model is usually run with static land cover and climatological vegetation leaf area index (LAI) throughout the simulation period as a result of specific model configuration (Wang et al., 2012; Xie et al., 2015). Previous studies have confirmed that the simulation accuracies of VIC model have been obviously improved in the intra-annual dynamics of soil moisture (Ford & Quiring, 2013), evapotranspiration (Tang et al, 2012) and runoff (Zhai et al., 2021) when remotely sensed intra-annual LAI dynamics instead of constant climatological LAI were used as input data during simulation process. However, vegetation phenological dynamics and LAI can show a large interannual variation (Wu et al., 2016; Piao et al., 2019), and VIC simulations considering year-to-year variability of LAI are able to better capture the interannual variation of runoff (Tesemma et al., 2015). Therefore, traditional configuration in land cover and vegetation parameters of VIC model probably underestimate the cumulative contribution of interannual vegetation change to the hydrological cycle (Xie et al., 2015). Improvement of the VIC model by coupling yearly land cover and continuously dynamic vegetation parameters that can be retrieved from remote sensing data sets would be favorable to remedy this issue (Tang et al., 2008, Xie et al., 2015; Yang et al, 2019).

Table 1 summarizes some typical studies about the attribution of annual runoff change in YRB and in China. Apparently, the inconsistencies between these studies stem from the different methods, time periods, and base scenarios used. None of these studies examined the influence of the temporally explicit vegetation change and climate variation on natural streamflow trend across the YRB. The specific objectives of this paper include to: 1) develop a VIC simulation scheme which enables VIC to reflect the cumulative effect of dynamic vegetation on the hydrological cycle by coupling time-series land cover and LAI remote sensing data; 2) assess the impacts of interannual change and intra-annual temporal pattern change of climatic factors, interannual change and intra-annual temporal pattern change of vegetation, and their interactive effect on the streamflow trend of YRB during 1982-2018; 3) compare the difference in attribution of streamflow change using VIC with and without considering continuous dynamics of LAI, and analyse the underlying causes of effects of different influencing factors on streamflow reduction.

**Table 1. Summarizing typical studies carried out in the YRB and China for attributing interannual streamflow change**

| Study | Region | Method | Purpose |
|---|---|---|---|



| | | | |
|---|---|---|---|
| Tang et al. (2008) | Yellow River Basin | Distributed biosphere hydrological (DBH) model | Assessing the impacts of interannual change and temporal pattern change of climatic factors, interannual vegetation change on the change trend of interannual streamflow during 1960-2000. |
| Gao et al. (2011) | The middle reaches of the Yellow River | Double mass curve | Separating the impacts of precipitation and human activities on the multi-year average change of streamflow between 1950-1985 and 1985-2008. |
| Tang et al. (2013) | Yellow River Basin | Soil and water assessment tool (SWAT) | Estimating the impacts of interannual change of climatic factors on the multi-year average change of streamflow between 1960-1990 and 2003-2011. |
| Cuo et al. (2013) | The source region of the Yellow River | Variable infiltration capacity (VIC) model | Assessing the impacts of interannual change of climatic factors and land cover change on the change trend of interannual streamflow during 1959-2009. |
| Xie et al. (2015) | Three-North region of China | Variable infiltration capacity (VIC) model | Assessing the impacts of interannual change of climatic factors and multi-year average change of vegetation on the change trend of interannual streamflow during 1989-2009. |
| Wang et al. (2017) | China | Snowmelt-based water balance model (SWBM) | Exploring the runoff sensitivity to climate change for hydro-climatically different catchments in China during 1956-2016 |
| Yang et al. (2019) | Loess Plateau of China | Variable infiltration capacity (VIC) model | Estimating the impacts of interannual change of climatic factors and multi-year average change of vegetation on the multi-year average change of streamflow between 1984-1999 and 2000-2015. |
| Wang et al. (2021) | The middle reaches of the Yellow River | Budyko-based elastic coefficient method | Assessing the impacts of multi-year average change of climatic factors and underlying surface condition on the multi-year average change of streamflow between 1956-1996 and 1997-2016 |
| Zhai et al. (2021) | China | Variable infiltration capacity (VIC) model | Assessing the impacts of interannual change of climatic factors and multi-year average change of vegetation on the multi-year average change of streamflow between 1982-1984 and 1982-2016. |
| This study | Yellow River Basin | Variable infiltration capacity (VIC) model | Assessing the impacts of **interannual change and intra-annual temporal pattern change** of climatic factors and vegetation, and the **interactive effect** of climatic factors and vegetation change on the change trend of interannual streamflow during 1982-2018. |

## 2 Study area and data

**2.1 Study area**

In the YRB, the area above the Tangnaihai (TNH) hydrologic station (100°09'E, 35°30'N) is defined as the headwater region. The Toudaoguai (TDG) station (111°04'E, 40°27'N) is the demarcation point between upper and middle reaches. The region between the TDG and Huayuankou (HYK) gauge (113°39'E, 34°55'N) is the middle reaches where the region between the TDG and Longmen (LM) gauge (110°35'E, 35°40'N) is the main sedimentation formation area of the YRB. The study area is

105 the catchment above the Huayuankou station with a drainage area of 730,036 km² (~97% of the total area of the YRB), and the mean annual runoff in the study area accounts for ~98 % of that in the whole YRB (Tang et al., 2013). Areas of contribution for TNH, TDG, LM, and HYK are approximately 121,972 km², 367,898 km², 497,552 km², and 730,036 km², respectively. The study area is divided into 4 sub-regions (Source region, TNH-TDG, TDG-LM, LM-HYK) between the target gauge and the adjacent upstream gauge from TNH gauge to HYK gauge, as illustrated in the Figure 1.

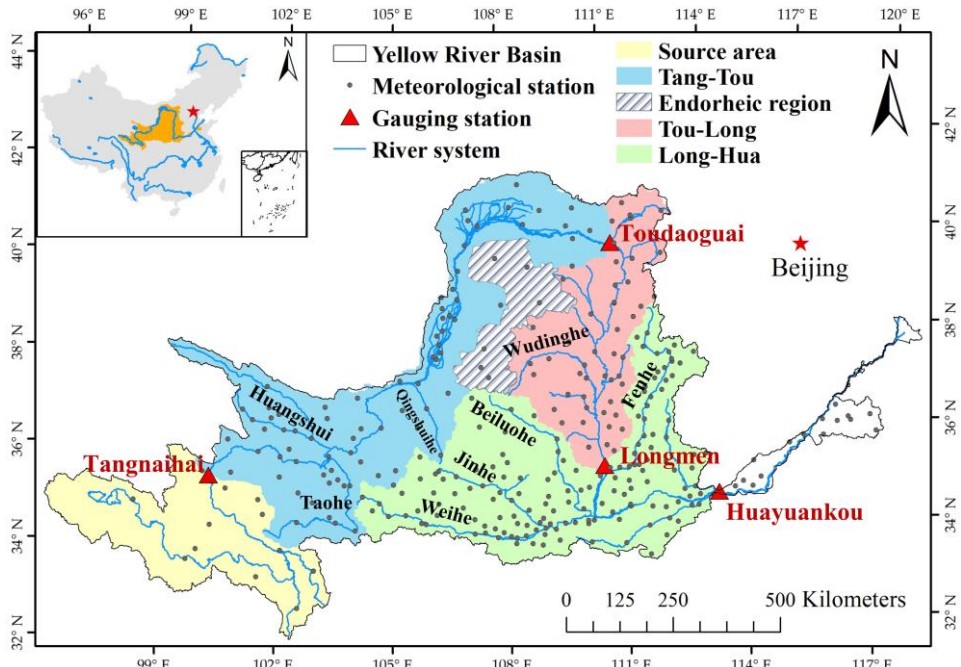

**Figure 1. Spatial distribution of the meteorological and streamflow gauge stations in the YRB. The insert map shows location of the YRB in China.**

## 2.2 Data sources

The observed daily data from 265 meteorological stations, including daily time series of precipitation, maximum temperature, minimum temperature, and wind speed from 1980 to 2018 were obtained from the China Meteorological Administration (http://data.cma.cn/). We calculated the daily mean temperature by averaging daily maximum and minimum temperature. The 8-days time series of vegetation leaf area index (LAI) at 500 m from 1982 to 2018 used in this study was obtained from The Global Land Surface Satellite (GLASS) product (Xiao et al., 2014) (http://glass-product.bnu.edu.cn/). We obtained land cover data for every 5 years during 1985-2020 from the GLC_FCS30 product (Zhang et al., 2021), which was the first global land cover product with fine classification system at 30 m (http://www.geodata.cn/). Elevation data obtained from the Shuttle Radar Topography Mission (SRTM) digital elevation dataset at 90 m (https://www.gscloud.cn/) was used to delineate river networks that are necessary for runoff routing of hydrological model. The soil texture data were derived from the 1-km China soil map based harmonized world soil database (HWSD) (v1.1) (http://data.tpdc.ac.cn/en/). The China terrace proportion map at 1 km resolution (Cao et al., 2021) in 2018 was download from https://doi.org/10.5281/zenodo.3895585. Global surface water product at 30 m from 1984 to 2020 was available from the Joint Research Centre (JRC) (https://global-surface-water.appspot.com/download).

For runoff, there are 4 mainstream gauges shown in Figure 1. Monthly naturalized runoff from 1980 to 2018 were provided by Yellow River Conservancy Commission of Ministry of Water Resources. Naturalized runoff at the target gauge was estimated by adding hman water use data from irrigation, industrial and domestic sectors over the drainage area of the



target gauge back to the observed runoff at the target gauge (Yuan et al., 2017; Zhang et al., 2020). We used naturalized runoff to calibrate hydrological model for simulating natural hydrological processes.

## 3 Methodology

### 3.1 Change detection of streamflow and influencing factors

We used the slope of the simple linear regression (Wang et al., 2022) to characterize the interannual change trend of
streamflow and influencing factors including precipitation, temperature, wind speed and LAI over the YRB. The t-test was used to examine the significance level of this trend. In addition, contribution of monthly streamflow change at a given month to the annual streamflow change was also determined by dividing the trend of monthly streamflow by the trend of annual streamflow.

Due to the changes of intra-annual temporal pattern in the precipitation and LAI are also able to affect the annual
streamflow, we taken the ratio of observed monthly to annual precipitation or LAI as the indicator of intra-annual temporal pattern in this study, and its change trend of each month was also analyzed. To explore more details on the relationship between temporal variability of precipitation and streamflow, double mass curve (Zhang et al., 2011) was performed to detect the abrupt change point and baseline period in the annual streamflow time series (Mu et al., 2007; Gao et al., 2010).

### 3.2 VIC  model setup considering temporally explicit vegetation change

The VIC model uses the variable infiltration curve (Liang et al., 1994) to account for the spatial heterogeneity of runoff generation. It assumes that surface runoff for the upper two soil layers is generated by those areas where precipitation exceeds the storage capacity of the soil. The methods from the ARNO model (Todini, 1996) were used to describe base flow generation which only happened in the third soil layer. A separate routing model was then coupled with the VIC model to simulate streamflow (Lohmann et al. 1998), where the runoff generated in each grid cell is routed to selected points through
the channel network.

To balance the high cost of computation and the characterization of heterogeneous underlying surface, we performed simulations using the VIC model on a 0.1°×0.1° grid scale at a daily timestep. The inputs of the VIC model include meteorological forcings, vegetation parameters, land cover and soil parameters. The meteorological forcings were derived by interpolating gauged daily precipitation, maximum and minimum temperature, and wind speed from stations into a
resolution of 90 m based on the AUSPLINE software and DEM data, and we then calculated the spatial average of interpolated data within a grid cell, as illustrated in the Figure 2. By default setting, VIC simulation only considers the climatology of vegetation (e.g., 12-month LAI), and the interannual change of LAI and land cover are time-invariant during the implementation of the VIC model. Therefore, the impacts of continuous interannual change of LAI and land cover types


on hydrological processes rarely be discussed in previous studies (Xie et al., 2015; Yang et al., 2019; Zhai et al., 2021). In
this study, the VIC model simulation scheme considering time-variant LAI was designed as the following two steps:

**Step I:** GLC_FCS30 product was firstly resampled to the same resolution (500 m) of LAI product. Owing to lack of yearly
land cover data, the land cover data in the ith year from GLC_FCS30 product was used to represent the land cover from (i-
4)th year to ith year. We smoothed the 8-days LAI time series with the adaptive Savitzky–Golay filter (Chen, et al. 2004) to
eliminate the abnormal LAI contaminated by cloud, signal errors from sensor, etc. The smoothed 8-days LAI was then
aggregated to a monthly value with temporal averaging for each year. Finally, the area fractions and average monthly LAI
value for each land cover type in each $0.1° \times 0.1°$ grid cell in each year were calculated respectively (Figure 2).

**Step II:** In the process of running VIC model, area fraction and monthly LAI for each land cover type in each grid cell in the
ith year were inputted into the VIC model, meanwhile the hydrological state at the last day in this year was saved. When
starting hydrological simulation in the (i+1)th year, the area fraction and monthly LAI for each land cover type in each grid
cell in the (i+1)th year and the hydrological state at the last day in the ith year were taken as the input data of VIC model.
This cycle running scheme demonstrated in the Figure 2 can enable VIC model to successfully simulate hydrological process
considering temporally explicit LAI and land cover change.

The soil physical parameters (e.g., field capacity, wilting point, and saturated hydraulic conductivity) are specified based
on the soil texture of HWSD and the algorithms introduced by Maurer et al. (2002). The soil parameters that were not
available from the HWSD were extracted from global soil datasets (Nijssen et al., 2001a). These soil data for VIC show
great advantages for retrieving global soil moisture (Nijssen et al., 2001b) and river discharges (Nijssen et al., 2001a). The
remaining numerical soil parameters were determined via model calibration following the method described in the section
3.3.

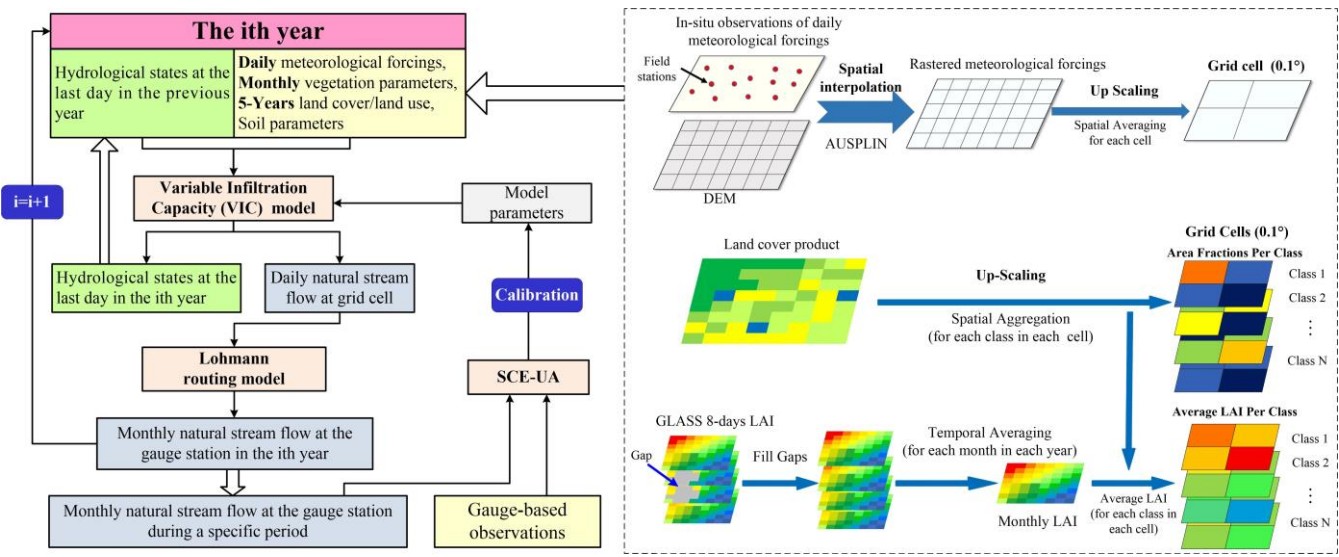

**Figure 2. The flowchart of VIC model setup considering temporally explicit vegetation change**





### 3.3 Model calibration and evaluation

The objective of this study was to investigate the contributions of changes in climate and vegetation to runoff changes, rather than to simulate runoff accurately from 1982 to 2018. Therefore, we adopted the baseline period to calibrate the 6
numerical soil parameters, including the infiltration parameter b, the second and third soil layer depths ($d_2$ and $d_3$), and the three parameters in the base flow scheme ($D_m$, $D_s$, and $W_s$) (Xie et al. 2007; Shi et al. 2008), in different sub-regions.

To find the optimal parameter set, an optimization algorithm of the multi-objective complex evolution of the University of Arizona (MOCOM-UA) from Yapo et al. (1998) was implemented, and Nash–Sutcliffe efficiency (NSE), average relative bias (Bias) and root mean square error (RMSE) were used as the objective function to assess the model performance. The
automatic calibration was carried out by running the VIC model thousands of times over each 0.1-degree grid cell within the YRB during calibration period, of which the first two years (1980–1981) used for warm up. This study assumes that the same amount of relative bias of annual streamflow trend during the calibration period will be transformed to the simulation period, and this relative bias was then deducted when calculating impacts of climate and vegetation on runoff (Luan et al., 2020). In this way, we can minimize the impact of hydrological simulation error in attributing annual streamflow change
trend.

### 3.4 Attributing the impacts of vegetation change and climate variation on streamflow trend

### 3.4.1 Reconstruction of de-trended climate variables and vegetation data

In this study, control conditions of climatic variables and LAI are defined as de-trended values rather than multi-year mean values adopted in other researches, because the interannual variability of the original time series can be preserved. The linear
trend of the variable at annual scale was removed according to the processing steps in the study of Xie et al. (2015), and a similar de-trended strategy was successfully used by Tang et al. (2008) and Bai et al. (2019) to examine the impacts of climate change and vegetation. Daily precipitation and monthly LAI time series required for VIC model were reconstructed using Eq.(4)-Eq.(5), as follows:

$$P_{daily} = \frac{P_{daily}}{P_{monthly}} \times \frac{P_{monthly}}{P_{annual}} \times P_{annual} \tag{4}$$

$$LAI_{monthly} = \frac{LAI_{monthly}}{LAI_{annual}} \times LAI_{annual} \tag{5}$$

Where, $P_{daily}$ is the daily precipitation time series, $P_{monthly}$ and $LAI_{monthly}$ are the monthly precipitation and LAI time series, $P_{annual}$ and $LAI_{annual}$ are the annual precipitation and LAI time series.

We generated the $P_{daily}$ time series where the trend of annual value was removed using de-trended $P_{annual}$ and original $\frac{P_{daily}}{P_{monthly}}$ and $\frac{P_{monthly}}{P_{annual}}$, and generated the $P_{daily}$ time series where the trends of both annual value and intra-annual temporal





pattern were removed using de-trended $P_{annual}$ and $\frac{P_{monthly}}{P_{annual}}$ and original $\frac{P_{daily}}{P_{monthly}}$. Likewise, de-trended monthly LAI time

series can be derived using same method.

### 3.4.2 Scenario simulation experiments

To explore the relative contributions of temporally explicit vegetation change and climate variation on annual streamflow
trend, we designed several scenario simulations (Table 2). We first simulated the interannual streamflow trend when
interannual change of annual values and intra-annual temporal pattern are de-trended for all climatic variables and LAI, and
the land cover is fixed at the year of 1982 (Scenario S1), thus representing the baseline scenario under the control condition
of unchanged climatic variable, vegetation and land cover during 1982-2018.

**Table 2. Scenario simulation experimental design to attribute the effects of climate change and vegetation change on the runoff trend.**

| Scenarios | Climate variables | | LAI and land cover | | Purposes |
|---|---|---|---|---|---|
| | Interannual change | Interannual change of intra-annual temporal pattern of rainfall | Interannual change | Interannual change of intra-annual temporal pattern of LAI | |
| **S1** | De-trended | De-trended | De-trended and fixed | De-trended | Estimating the runoff without any climate change and vegetation change |
| **S2** | Observed | De-trended | De-trended and fixed | De-trended | Estimating the impact of interannual change of climate variables |
| **S3** | Observed | Observed | De-trended and fixed | De-trended | Estimating the impact of intra-annual temporal pattern change of climate variables |
| **S4** | De-trended | De-trended | Observed | De-trended | Estimating the impact of interannual change of vegetation |
| **S5** | De-trended | De-trended | Observed | Observed | Estimating the impact of intra-annual temporal pattern change of vegetation |
| **S6** | Observed | Observed | Observed | Observed | Estimating the interactive effect of climatic factors and vegetation change |

To isolate the effect of climate variables on streamflow trend, we designed two scenarios. In Scenario S2, annual value of
specific climate variable varied according to observation records while other variables vary according to control conditions
in the S1. In Scenario S3, annual values of all climate variables and intra-annual temporal pattern of monthly precipitation
vary according to observation records while other variables vary according to control conditions in the S1. The impacts of
climate variables were calculated as follows:

$$Q_{C_{inter}} = f(C_{inter}) - f(control) \tag{6}$$

$$Q_{P_{intra}} = f(C_{inter}, P_{intra}) - f(C_{inter}) \tag{7}$$

$$Q_C = Q_{C_{inter}} + Q_{P_{intra}} = f(C_{inter}, P_{intra}) - f(control) \tag{8}$$





Where $Q_{C_{inter}}$ and $Q_{P_{intra}}$ are impacts of interannual change of each climate variable and intra-annual temporal pattern of precipitation on the annual streamflow trend. $Q_C$ represents the total impacts of all climate variables. $f(control)$, $f(C_{inter})$ and $f(C_{inter}, P_{intra})$ are the simulated streamflow trends in the S1, S2 and S3, respectively.

To isolate the effect of vegetation on streamflow trend, we designed two more scenarios. In Scenario S4, annual values of LAI and land cover vary according to remote sensing observation records while other variables vary according to control conditions in the S1, and both annual values of LAI and land cover and intra-annual temporal pattern of monthly LAI vary according to observation records while all climatic variables are de-trended in Scenario S5. The impacts of vegetation were calculated as follows:

$$Q_{LAI_{inter}} = f(LAI_{inter}) - f(control) \qquad (9)$$

$$Q_{LAI_{intra}} = f(LAI_{inter}, LAI_{intra}) - f(LAI_{inter}) \qquad (10)$$

$$Q_{LAI} = Q_{LAI_{inter}} + Q_{LAI_{intra}} = f(LAI_{inter}, LAI_{intra}) - f(control) \qquad (11)$$

Where $Q_{LAI_{inter}}$ and $Q_{LAI_{intra}}$ are impacts of interannual change of annual values and intra-annual temporal pattern of vegetation on the annual streamflow trend. $Q_{LAI}$ represents the total impacts of vegetation. $f(LAI_{inter})$ and $f(LAI_{inter}, LAI_{intra})$ are the simulated streamflow trends in the S4 and S5, respectively.

To identify the interactive effect of climate variables and vegetation on streamflow trend, we additionally designed the Scenario S6 to simulated the actual trend of streamflow based on dynamic climate variables and vegetation from 1982 to 2018, thus representing the combined effects from both climate and vegetation. Due to interactive effects of predictor variables on response variable can be interpreted as the second-order or higher-order terms in Multi-point Taylor expansion (Bai et al., 2019), the interactive effect of climate variables and vegetation can be derived as follows:

$$Q_{C \times LAI} = f(C, LAI) - f(control) - Q_C - Q_{LAI} \qquad (12)$$

Where $Q_{C \times LAI}$ is the interactive effect of climatic factors and vegetation on the annual streamflow trend. $f(C, LAI)$ is the simulated streamflow trend in the S6.

The impact of residual factors (e.g. non-vegetation underlying surface) was calculated by the residual method, as illustrated in the Eq.(13). The relative impact rate of each influencing factor on the annual streamflow trend were calculated using the Eq. (14).

$$Q_{Resi.} = Q_{nat} - Q_C - Q_{LAI} - Q_{C \times LAI} \qquad (13)$$

$$Contr. X_i = \frac{Q_{X_i}}{\sum_{i=1}^{n} |Q_{X_i}|} \times 100\% \qquad (14)$$

Where $Q_{nat}$ is the change trend of naturalized streamflow. $Q_{Resi.}$ is the impact of residual factors on the annual streamflow trend. $Q_{X_i}$ and $Contr. X_i$ are the impact and relative impact rate of ith (i=1,2,…,8) influencing factor respectively. The positive $Contr. X_i$ represents the positive impact to the streamflow change, and vice versa.



# 4 Results

## 4.1 Annual natural streamflow trend over YRB

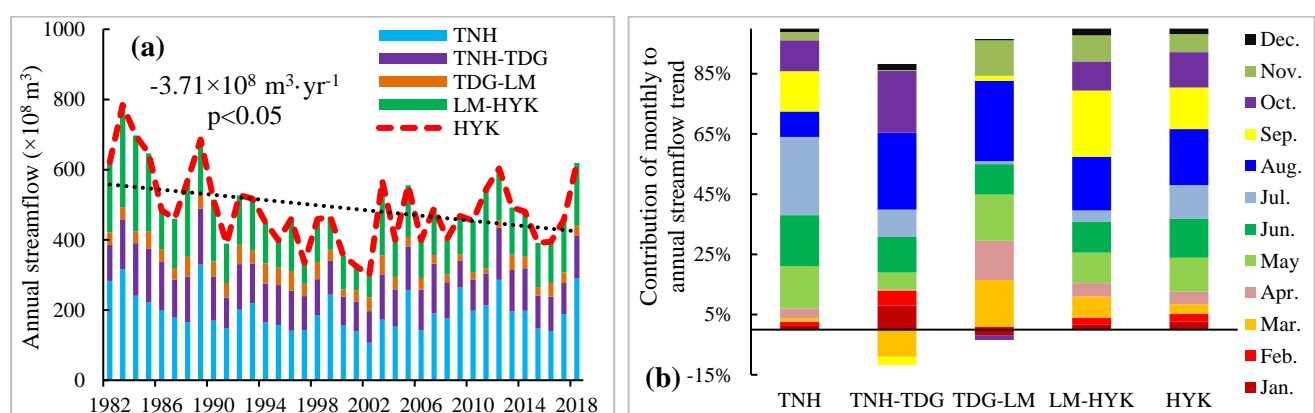

**Figure 3. (a) Naturalized annual streamflows of HYK and different sub-regions, (b) contributions of monthly to annual streamflow trend of HYK and different sub-regions.**

The Naturalized annual streamflows of HYK station, source region, TNH-TDG, TDG-LM and LM-HYK were provided in Figure 3(a). A significant decreasing trend was observed from the annual streamflow time series of HYK station during 1982-2018, with a negative trend of $-3.71 \times 10^8$ m$^3 \cdot$yr$^{-1}$. Spatially, all sub-regions reported downward annual streamflow trends, with different contributions of 20.7%, 20.6%, 14.6% and 44% on the annual streamflow trend of HYK from source region to LM-HYK. Temporally, all monthly streamflow experienced negative trends at HKY station, with a greatest reduction (18.6%) was found in August. Most monthly streamflow trends of four sub-regions were negative, and the greatest contributions of monthly trend to annual trend occurred in the July for source region, August for TNH-TDG and TDG-LM, and September for LM-HYK.

## 4.2 Temporally explicit change of climatic factors and vegetation

### 4.2.1 Interannual trend of climatic factors and LAI

The spatio-temporal change characteristics of interannual climate variables and LAI time series were investigated based on the linear slope analysis, as illustrated in the Figure 4, where interannual variability of region-averaged value and percentage of area with different significance level were summarized. The YRB experienced insignificantly positive trend in annual precipitation, with significant increases in only 4.6% of the basin, and areas with decreasing trend were mainly located at Huangshui basin and southeast in the LM-HYK. In the context of global warming, 97.5% of the YRB exhibited a significant increasing trend in annual mean temperature, with a change rate of 0.07°C·yr$^{-1}$. In contrast, significant decreasing trends in annual mean wind speed occurred over 78.2% of the YRB, while Taohe and Weihe basins had slight upward trends. For annual mean LAI, most of the YRB (72.5%) experienced a significant increasing trend, especially for the LM-HYK. The downward LAI trend occurred in only 15% of the study area, which was caused by the vegetation degradation in the source





region and urbanization in the middle reaches. A sharp increase of LAI trend in the TNH-TDG, TDG-LM and LM-HYK were noted after the year of 2000 associated with the implementation of Grain for Green Project (GFGP).

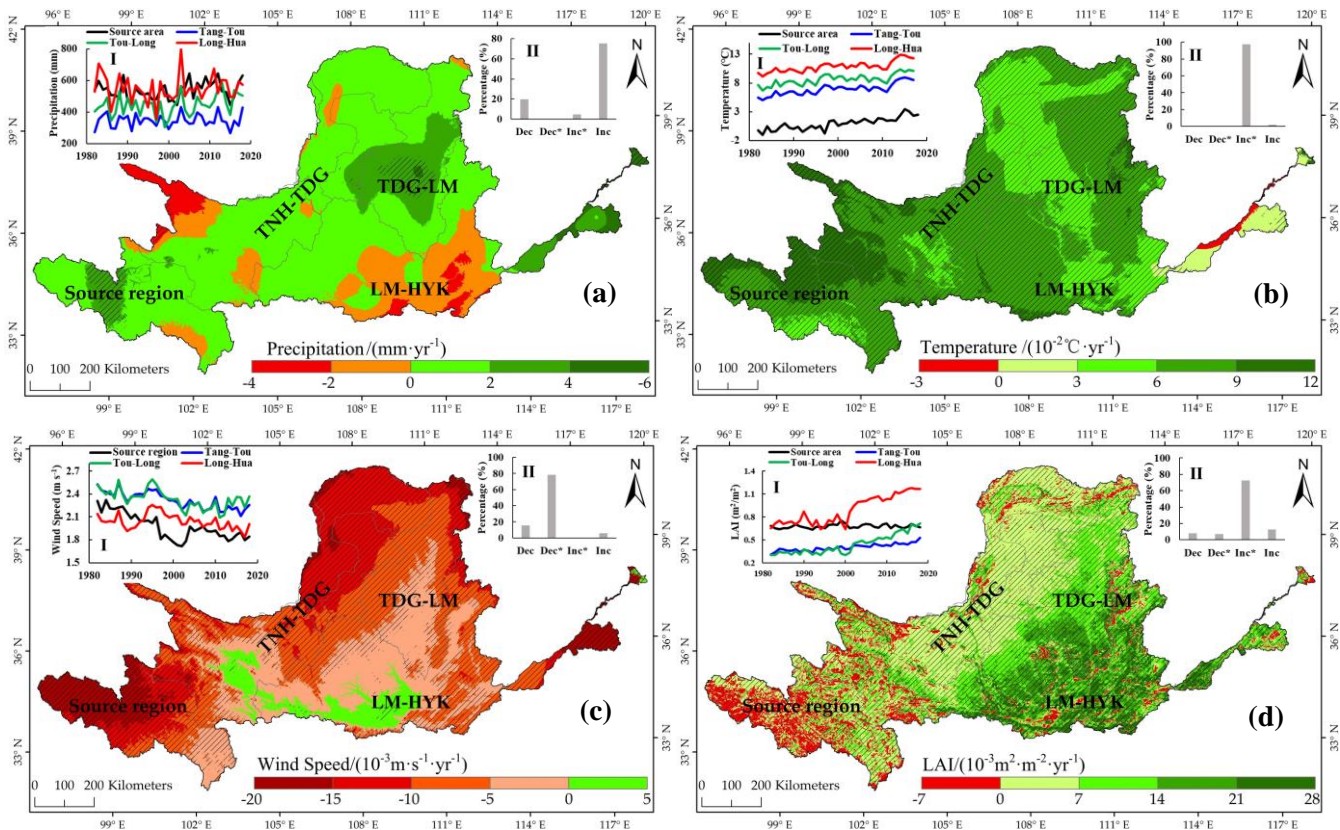

**Figure 4. The spatio-temporal change of in precipitation (a), temperature (b), wind speed (c) and LAI (d). The insets (I) show the interannual variation of region-averaged variables. The insets (II) show the percentages (%) of the area with significant decrease (Dec\*, p<0.05), insignificant decrease (Dec), insignificant increases (Inc), and significant increase (Inc\*, p<0.05).**

### 4.2.2 Interannual trend of intra-annual temporal pattern for precipitation and LAI

The statistics on the trends of monthly to annual precipitation ratio for four sub-regions were shown in the Figure 5. Negative trends primarily occurred between March and July, with July exhibiting the largest negative trends for all sub-regions except source region where a largest negative trend occurred in the June. Positive trends predominantly occurred between August and December, and September corresponded to the largest positive trends for all sub-regions except source region where a largest trend was observed in the August. It was indicated that the intra-annual temporal distribution of monthly precipitation has varied from 1982 to 2018, and positive contribution from the autumn season to annual precipitation has been progressively on the rise, whereas the contribution of summer has declined. Rainfall frequency caused by temporal pattern change of precipitation possibly influence the hydrological process over the YRB.





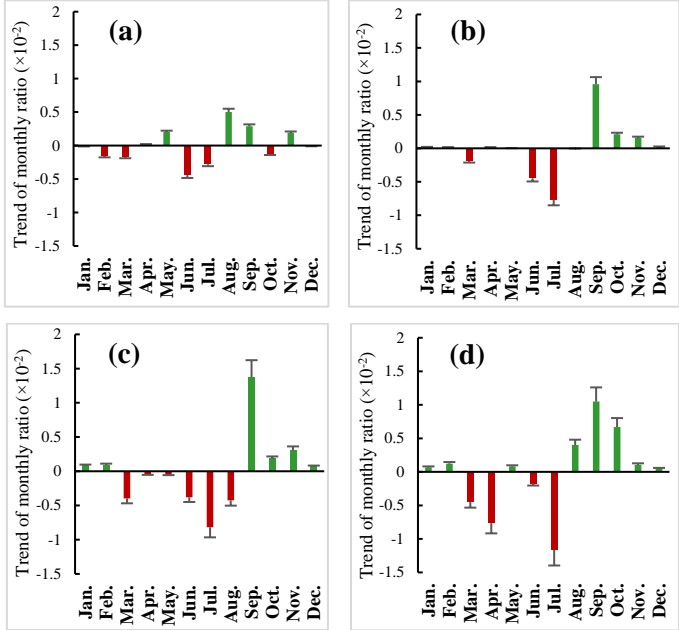

**Figure 5. Trends in the ratio of the observed monthly to annual precipitation of subregions in (a) souce region, (b) TNH-TDG, (c) TDG-LM, (d) LM-HYK. The error bars represent the one standard deviation (s.d.).**

The trends in the ratios of monthly to annual mean LAI for four sub-regions were shown in the Figure 6. Negative trends primarily occurred between June and September, whereas positive trends predominantly occurred remaining months. It was obviously observed that the intra-annual temporal pattern of monthly LAI has also been varying during 1982-2018. Comparing with the upper reaches, the temporal pattern change was relatively great in the middle reaches, where positive contribution from the spring season to annual LAI has increased. Intra-annual change of evapotranspiration and soil moisture induced by temporal pattern change of LAI would influence the hydrological process over the YRB.

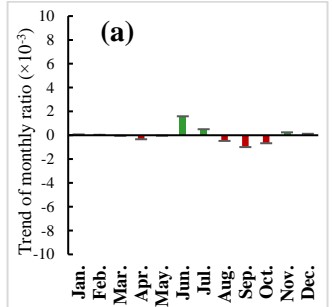

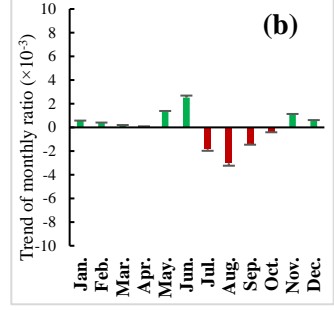





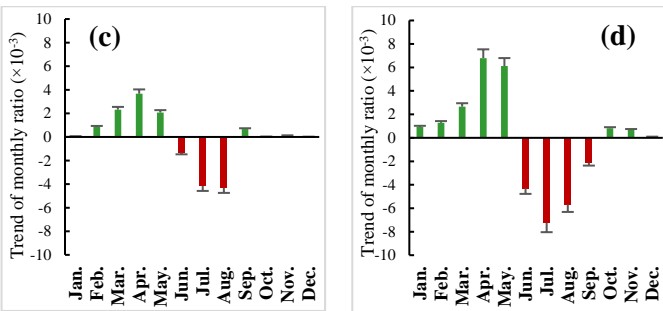

**Figure 6. Trends in the ratio of the observed monthly to annual mean LAI of subregions in (a) souce region, (b) TNH-TDG, (c) TDG-LM, (d) LM-HYK. The error bars represent the 1 s.d.**

## 4.3 Non-stationary relationship between precipitation and streamflow

**Figure 7. (a) The interannual change trend of annual runoff coefficiencts for 4 sub-regions, (b) precipitation-streamflow double mass curves for 4 sub-regions, and (c) precipitation-streamflow relationships in the two periods of 1982-1999 and 2000-2018 for 4 sub-regions.**





Runoff coefficients characterizing runoff yield capacity were calculated by streamflow divided by precipitation for all sub-regions, as illustrated in the Figure 7(a). It was found that overall trends in all sub-regions were negative during 1982-2018, although there were short-period upward trends from 2000 to 2018 in all sub-reigons excluding TDG-LM. To detect the abrupt change time of the realtionship between precipitation and streamflow, the cumulative curves of precipitation and streamflow of 4 sub-regions were calculated and plotted (Figure 7(b)). The discrepancy of cumulative precipitation and

streamflow observed from the Figure 7(b) indicated that the stationary precipitation-streamflow relationships have changed in all sub-regions, and the deviation of the streamflow from precipitation is more significant in the middle reaches than upper reaches. It is seen that significant abrupt changes in different sub-regions occurred in the same year of 1999. Thus, the study period was divided into two periods: 1982-1999 and 2000-2018. It is clearly seen that from the Figure 7(c) the precipitation-runoff relationship has significantly changed between these two periods, and the regression line of precipitation and runoff

during 1982-1999 always is above that during 2000-2018, which suggested that runoff in the period of 2000-2018 was significantly reduced when same precipitation in the period of 1982-1999 occurred. Therefore, it is reasonable to split the whole period into these two short-period. It could be concluded that the relationship between the annual precipitation and streamflow presents a non-stationary state in the YRB from 1982 to 2018.

### 4.4 Model evaluation

According to the calculated abrupt change point from the precipitation-streamflow double mass curves in Figure 7, the period of 1982-1999 was defined as the reference period, of which calibration and validation periods for calibrating VIC parameters were 1982-1993 and 1994-1999 respectively. The monthly hydrographs and average seasonal cycles of the simulated and naturalized streamflows for different catchment regions are shown in the Figure 8, respectively. According to the VIC simulations at HYK, the monthly NSE, RMSE and Bias are 0.89, 387.4 mm and -1.6% for the calibration period and

0.8, 386.6 mm and 6.9% for the validation period (Figure 8). Averaged across all four catchment regions, monthly NSE is 0.69, RMSE is 171.9 mm, and Bias is 5% during calibration period, and monthly NSE is 0.6, RMSE is 156.8 mm, and Bias is 9.5% during validation period (Figure 8). As per performance criteria given by Moriasi et al. (2007) (Table 3), simulation results indicate that the VIC model has a good performance in simulating hydrological processes in not only subbasins and sub-regions. In addition, Figure 8 also shows the multi-year average monthly streamflow during 1982-1999, and NSE is

larger than 0.85 in all catchment regions, except for TDG-LM, thus indicating the seasonal cycles of streamflow also can be perfectly captured by VIC model simulation.


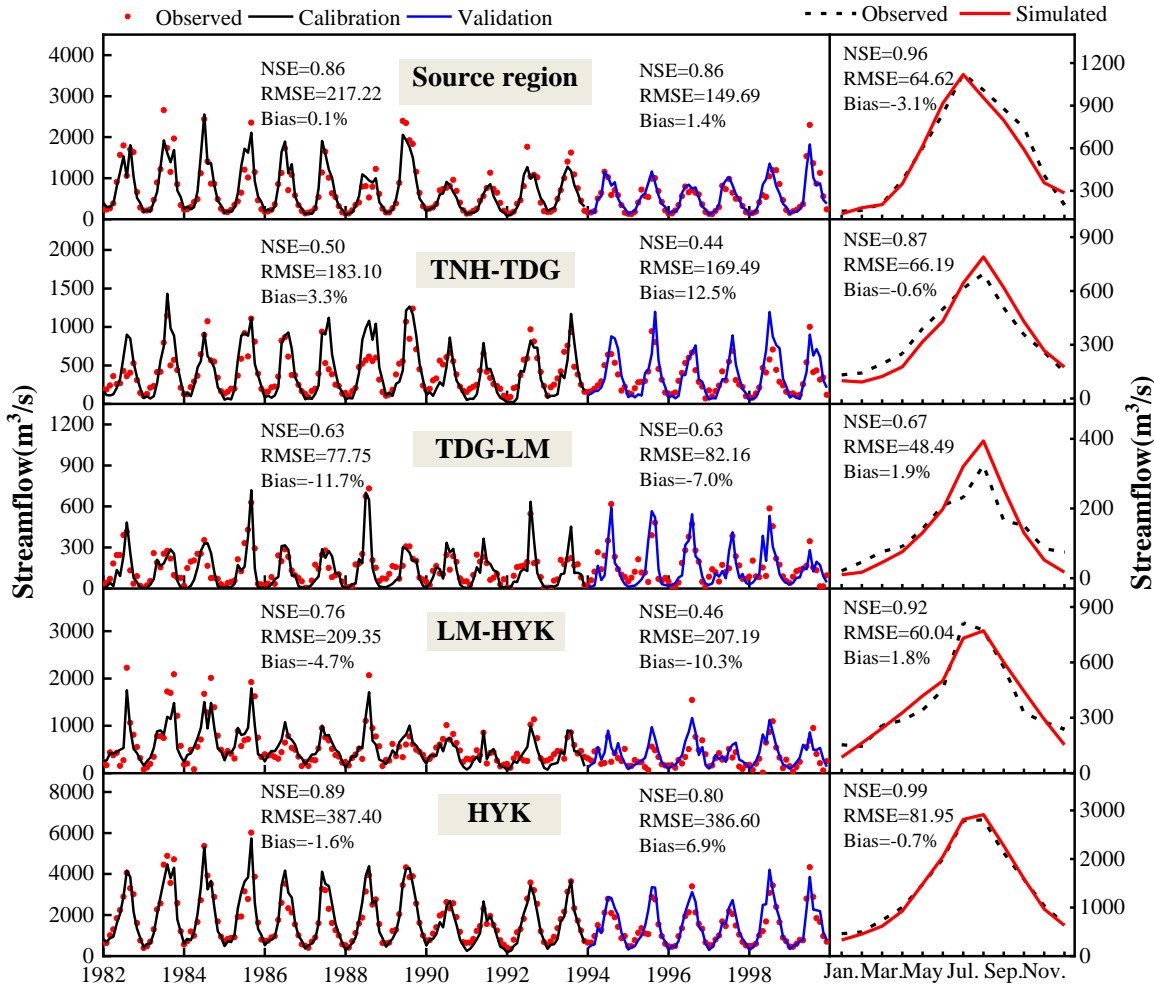

**Figure 8.** Comparisons of monthly streamflow and seasonal cycles of the VIC simulation and naturalized streamflow for different drainage areas during 1982-1999.

## 4.5 Impacts of influencing factors on the streamflow trend

The impacts and relative impact rates of eight influencing factors on the annual streamflow trends in different drainage areas were calculated using Eq.(6)-Eq.(14), as illustrated in the Figure 9. For the HYK station, the contributions of all climate variables to the streamflow trend were positive excepting temperature, while larger negative effects from underlying surface change offset the slight positive effects of climate change on the streamflow trend (Figure 9). From 1982 to 2018, the annual streamflow trend at HYK was $-3.71×10^8$ $m^3·yr^{-1}$, of which changes in interannual precipitation (P_inter), temperature (T_inter), wind speed (WS_inter), intra-annual temporal pattern of precipitation (P_intra), interannual LAI (LAI_inter), intra-annual temporal pattern of LAI (LAI_intra), interactive effects of climate variables and vegetation (Interactive), and residual underlying surface (Resi.) accounted for 15.1% ($1.14×10^8$ $m^3·yr^{-1}$), -23.5% ($-1.77×10^8$ $m^3·yr^{-1}$), 8.7% ($0.66×10^8$





m³·yr⁻¹), 1.4% (0.1×10⁸ m³·yr⁻¹), -26.6% (-1.99×10⁸ m³·yr⁻¹) and -6% (-0.45×10⁸ m³·yr-1), -3.5% (-0.26×10⁸ m³·yr⁻¹), -15.2%

(-1.14×10⁸ m³·yr⁻¹), respectively. It is concluded that vegetation change was the dominant driving factor for the long-term

decreasing trend of streamflow from 1982 to 2018 in YRB, meanwhile the effects in non-vegetation underlying surface

changes (e.g. water and soil conservation engineering measures, permafrost melting, etc.) on reducing streamflow cannot be

ignored.

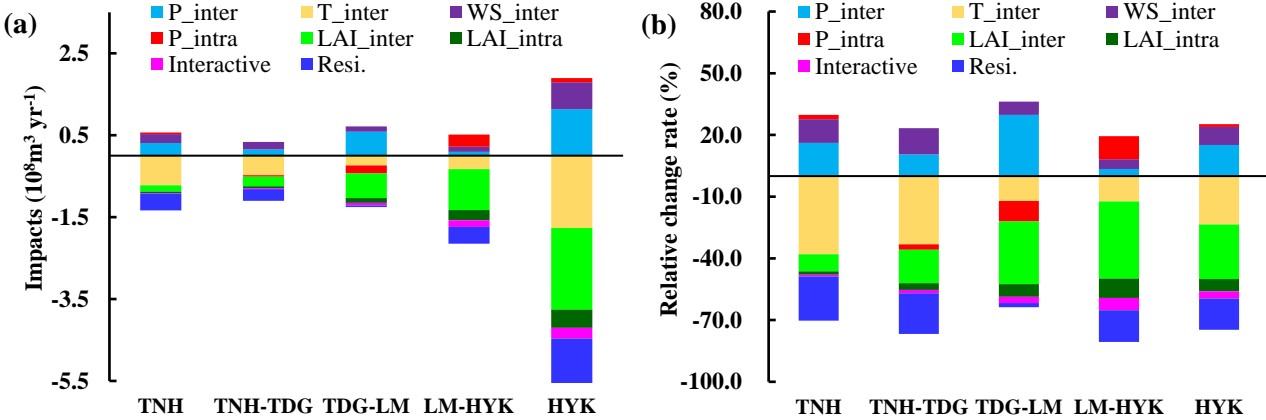


**Figure 9. (a) Impacts and (b) relative impact rates of the different influencing factors on the annual streamflow trends in different drainage areas over YRB.**

Due to divergent change of climate variables and underlying surfaces, the impact of different influencing factors on the

streamflow trend in different sub-regions exhibited obvious spatial variability. Net total effect from interannual changes of

all climate variables exhibited a negative influence on streamflow increase for all sub-regions, except for the TDG-LM with

a positive impact of 0.47×10⁸ m³·yr⁻¹. The contribution of temperature on decreasing trend of streamflow in the upper

reaches is greater than that in the middle reaches. Contributions of intra-annual temporal pattern change of precipitation on

the streamflow trend illustrated obvious spatial heterogeneities. The impact of this factor was positive in the source region

(2.1%) and LM-HYK (11.2%), whereas its negative effects were observed for the TNH-TDG (-2.6%) and TDG-LM (-9.9%).

It was found that not only interannual increase of LAI, but also intra-annual LAI temporal pattern change had effects of

reducing streamflow. Direct total impacts from vegetation change were negative for streamflow trend and accounted for -10%

(-0.19×10⁸ m³·yr⁻¹), -19.5% (-0.28×10⁸ m³·yr⁻¹), -36.6% (-0.72×10⁸ m³·yr⁻¹), and -46.9% (-1.25×10⁸ m³·yr⁻¹) of the

streamflow trends in source region, TNH-TDG, TDG-LM, and LM-HYK, respectively. Compared with direct effects of

vegetation change, the two-way interactive effects of vegetation and climate variables were relatively low in all sub-regions.

The impacts of residual underlying surface change were comparable to that of vegetation greening, with a maximum

contribution of -21.4% (-0.41×10⁸ m³·yr⁻¹) occurred in the source region.





## 5 Discussion

### 5.1 Impacts of temporally explicit precipitation change on the precipitation intensity

Previous studies have suggested that precipitation is the main factor controlling runoff change with climate change (Dan et
al., 2012; Wang et al., 2016; Liu et al., 2017). In this study, we further found that simulated annual streamflow trend could
be changed by not only interannual precipitation (S2-S1) but also intra-annual monthly to annual precipitation ratio (S3-S2),
which indicated that same annual precipitation with different intra-annual temporal pattern indeed affected the runoff
generation process (Tang et al, 2008). Due to runoff yield in excess of infiltration is the dominant runoff mechanism where
rainfall intensity is the crucial driving force over the most of YRB (Jin et al., 2020), we then focused on the impacts of
interannual precipitation and intra-annual monthly to annual precipitation ratio on the rainfall intensity.

Different precipitation intensities, including light, moderate, heavy precipitation, are defined as daily precipitation amounts
greater than 1, 10 and 25 mm, respectively in this region (Liu et al., 2018), and previous studies have proven that runoff was
more sensitive to total amount of heavy precipitation ($P_{25}$) by analysing a large number of in-situ observation data (Liu et al.,
2020). Therefore, the differences of interannual trends of $P_{25}$ between scenario S2 and S1 were calculated for each
meteorological station to indicate the impact of interannual precipitation on the heavy precipitation, as demonstrated in the
Figure 10 (a). The meteorological stations with an increasing trend in $P_{25}$ driven by interannual precipitation change
accounted for 69.7%, with a maximum proportion of 80% in the TDG-LM, which caused the increase of annual streamflow
(Figure 9).

Likewise, the impacts of intra-annual monthly to annual precipitation ratio on the $P_{25}$ were analysed using the combination of
scenario S3 and S2, as shown in the Figure 10 (b). The meteorological stations with an increasing trend in $P_{25}$ driven only by
intra-annual precipitation temporal patten change accounted for 58.9%, hence the overall effect of intra-annual temporal
pattern change on the naturalized streamflow was positive during the study period. Spatially, increasing trends of $P_{25}$ were
observed in the majority of the stations within the source region (60%) and LM-HYK (68%), whereas the decreasing trend
was dominant over the TNH-TDG and TDG-LM, which led to the spatial heterogeneity of the effects of intra-annual
precipitation temporal patten change (Figure 9).





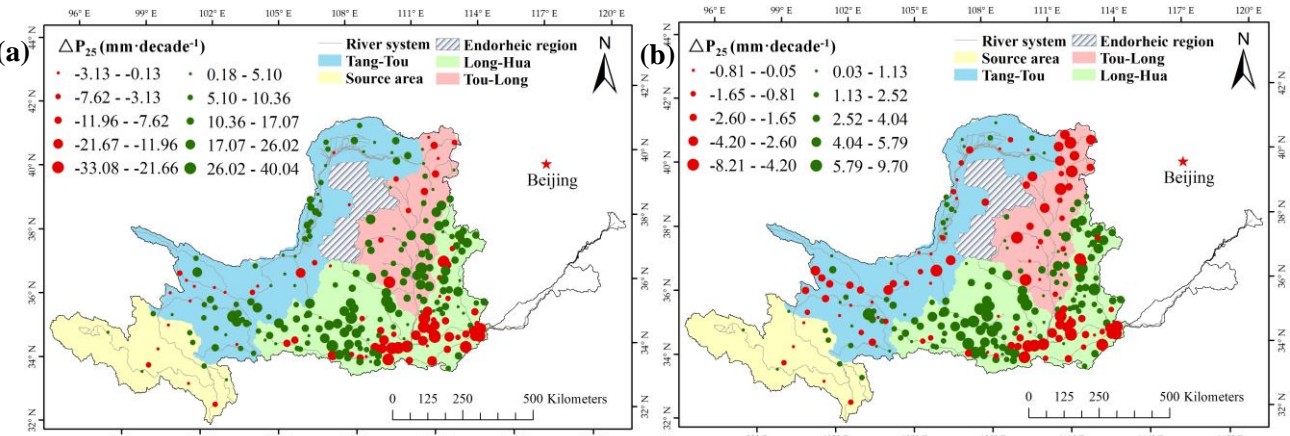

**Figure 10. The impacts of interannual precipitation (a) and intra-annual monthly to annual precipitation ratio changes (b) on the P25 trend over the YRB.**

## 5.2 Potential driving mechanisms of temporally change of LAI

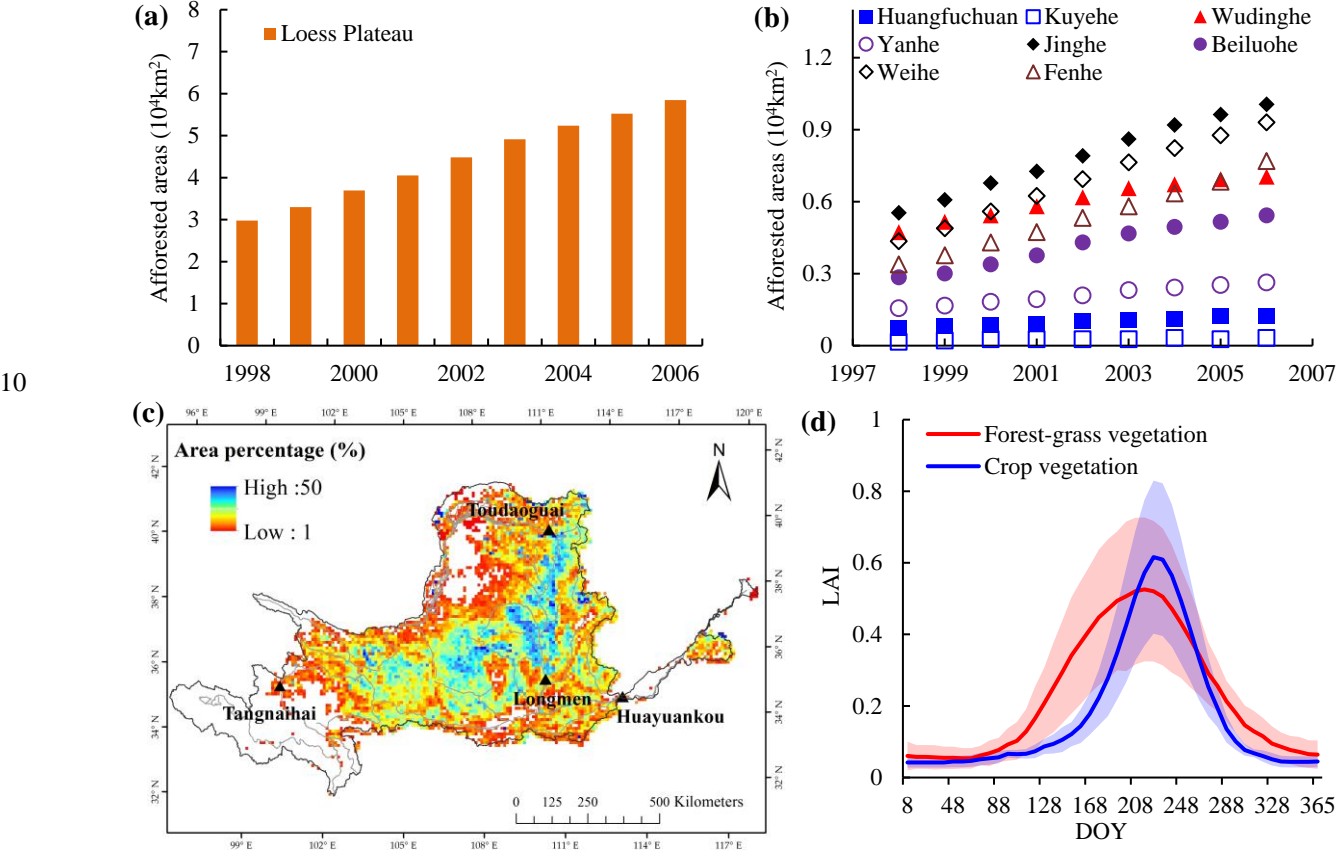






**Figure 11. (a) Total afforested areas implemented in the Loess Plateau from 1998 to 2006; (b) Afforested areas in different watersheds between 1998 and 2006; (c) spatial distribution of area percentage of the conversion of cropland into forest-grass during 1985-2020 in each 0.1° grid cell; (d) The intra-annual variation of LAI at 8-days scale for typical forest-grass vegetation and crop vegetation, and the solid line and shaded area indicate the mean and ±1 s.d.**


To mitigate increasingly devastating ecological environment and soil erosion problems, the Grain for Green Project (GTGP), which targets to convert farmland into forests and grasslands (Jia et al., 2014; Liu et al., 2014), have been implemented over the upper and middle reaches of the YRB since 1998. According to the statistical data from local forestry authorities (Yao et al., 2011), afforestation in the Loess Plateau has been mainly implemented during 1998-2006, and the

afforested areas across the plateau increased greatly from $3 \times 10^4$ km$^2$ in 1998 to $5.9 \times 10^4$ km$^2$ in 2006 (Figure 11 (a)). Between 1998 and 2006, artificially planted trees and shrubs rapidly increased, and the afforested areas of the Fenhe, Weihe, Beiluohe, Jinghe, Huangfuchuan, Yanhe, Kuyehe and Wudinghe watersheds increased by 128%, 113%, 93%, 82%, 76%, 66%, 55% and 49%, respectively, as shown in the Figure 11(b). Previous studies on the Loess Plateau have suggested that compared with climate change, the tree and grass planting activities was the dominant driving factor for the vegetation

greening (Sun et al., 2015; Zhang et al., 2016; Bai et al, 2019). In addition, natural rehabilitation without intensive interference activities, such as grazing prohibition, may play an important role in vegetation restoration in the Loess Plateau (Cao et al., 2011).

To explore the vegetation type conversion caused by GTGP, area percentage of the conversion of cropland into forest-grass during study period for each 0.1° grid cell was calculated using the GLC_FCS30. Figure 11(c) shows that massive

vegetation type conversion occurred in the TNH-HYK with a maximum percentage of 50%, which is partly proven by intense vegetation type conversion detected using Landsat time-series in the study of Wang et al. (2018). The Figure 11(d) depicts the phenological characteristics of typical crop vegetation and the forest-grass vegetation. The LAI of forest-grass vegetation in the spring and autumn season is obviously higher than that of farm crops, whereas LAI of crop vegetation in the summer season is slightly higher than that of forest-grass vegetation. Therefore, the massive vegetation type conversion

from cropland into forest-grass vegetation could significantly alter the vegetation phenological on the Loess Plateau, which could lead to the interannual trend of intra-annual monthly to annual LAI ratio increased in the spring and decreased in the summer (Figure 6). Meanwhile, due to phenology determines the start and end time of vegetation growth and is highly sensitive to climate change (Liang and Schwartz, 2009; Fu et al., 2019), climate warming has played an important role in advancing the spring phenology and delaying autumn phenology, and consequently extended the length of vegetation

growing period across the globe (Piao et al., 2019; Menzel et al., 2020), especially for the semi-arid and semi-humid regions of China(Wu et al., 2015; Chen et al., 2022).

## 5.3 Implication of considering temporally explicit vegetation change on hydrological effect assessment using VIC

In general, previous studies evaluated the hydrological effects of vegetation change using VIC model based on multi-year average LAI and vegetation types during different periods (Xie et al., 2015; Yang et al., 2019; Zhai et al, 2021) as a result of





the model configurations of VIC (Liang et al., 1994; Xie et al., 2007). However, due to the smoothing effect of averaging, multi-year average LAI is unable to fully capture the spatiotemporal vegetation change, especially for the area with tremendous ecological restoration. Therefore, to explore the discrepancy in evaluating the hydrological effect of vegetation using VIC considering and without considering temporally explicit LAI change, we calculated the annual streamflow trend change by differencing simulation of scenario S1 and simulation with dynamic annual LAI observations while other

variables varied under control conditions in the S1, and then calculated the streamflow trend change using the combination of scenario S1 and simulation where annual LAI during 1982-1999 and 2000-2018 were fixed into the multi-year averages of corresponding periods respectively, while other variables varied same as S1. Likewise, the annual streamflow trend changes simulated by continuous and noncontinuous change of intra-annual temporal pattern of LAI were also calculated using same way.

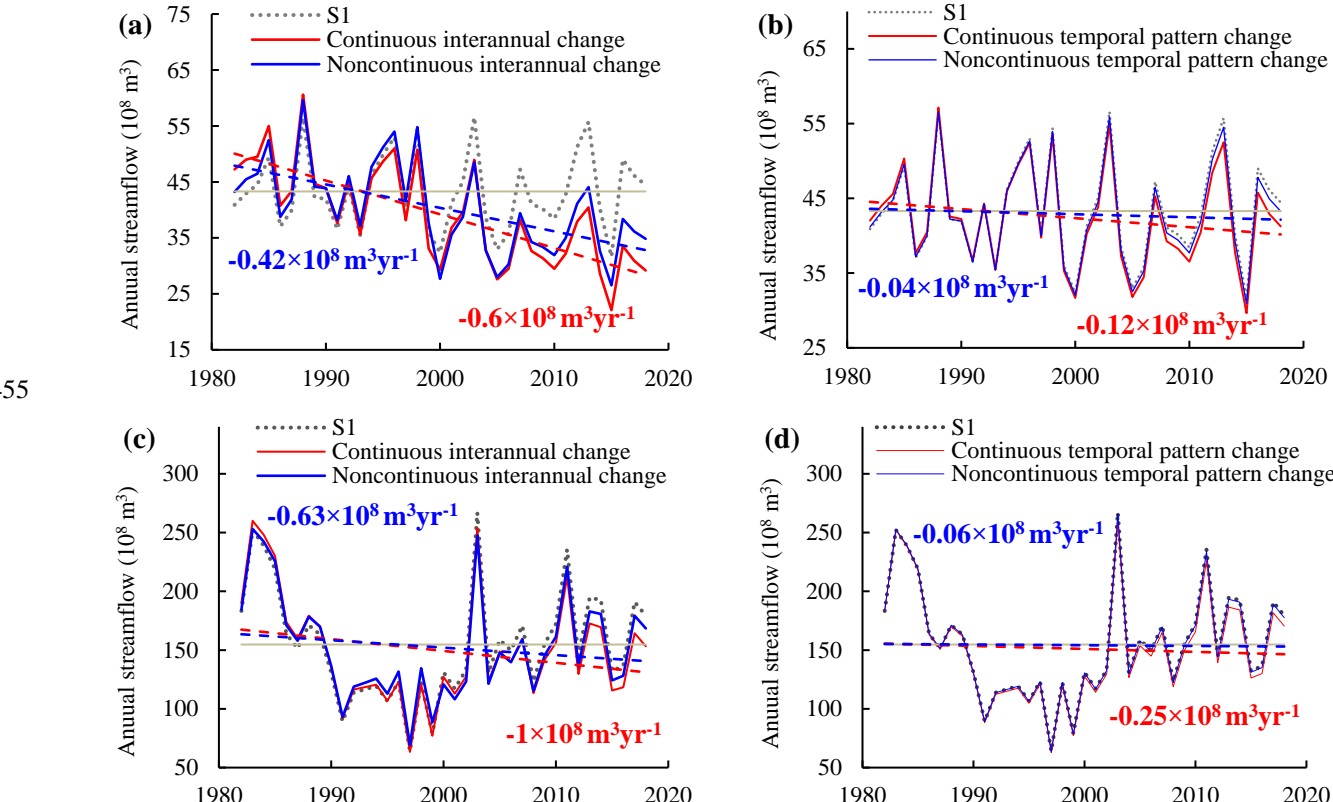

**Figure 12. The comparison of simulated annual streamflow trend using VIC considering and without considering continuous dynamics of interannual LAI and intra-annual temporal pattern of LAI for the TDG-LM (a~b) and for the LM-TDG (c~d).**

Figure 12 shows the comparison of simulated annual streamflow trend using VIC considering and without considering
continuous dynamics of interannual LAI and intra-annual temporal pattern of LAI in the TDG-LM (Figure 12(a)~(b)) and LM-HYK (Figure 12(c)~(d)). It is found that compared with simulation with multi-year average LAIs change, the impact of vegetation simulated by continuous LAI change was increased by 42.9% and 58.7% for TDG-LM and LM-HYK



respectively, and the impact of vegetation simulated by continuous intra-annual temporal pattern change was 3 times and 4.2 times of that simulated by noncontinuous inputs for TDG-LM and LM-HYK respectively. These results were consistent with

the reported attribution of runoff change in the upland Mediterranean basin where reductions in runoff were less intense when afforestation was not considered in the hydrological model (Buendia et al., 2015).

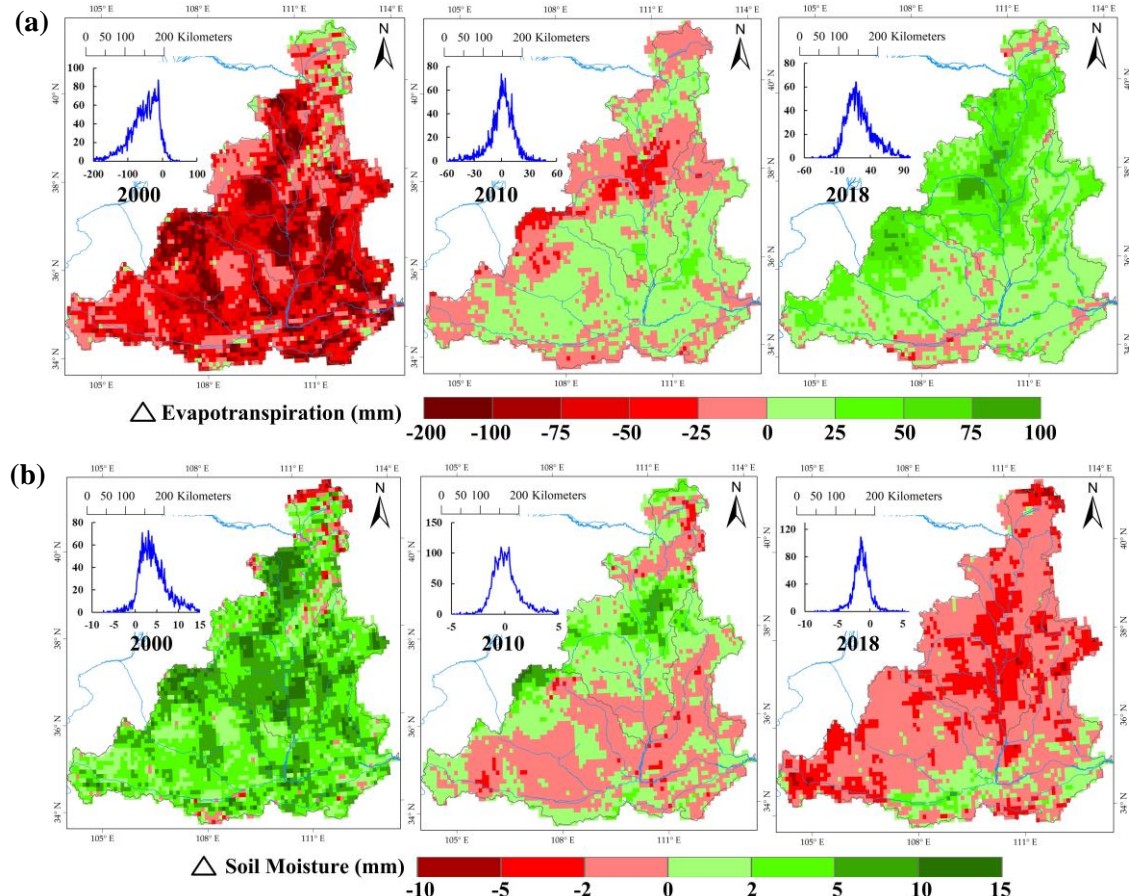

**Figure 13. The difference between two simulations by VIC with dynamic LAI and fixed multi-year average LAI during 2000-2018**
**for annual total evapotranspiration (a) and annual average soil moisture (b) in the middle reaches in the year of 2000, 2010 and**
**2018.**

Previous studies focusing on this region at basin scale or regional scale have confirmed that massive vegetation greening has increased regional evapotranspiration through intense transpiration and canopy interception (Feng et al. 2016; Shao et al., 2019; Bai et al., 2019; Li et al., 2020), and caused a dried layer in the soil profile, interfering the vertical infiltration of soil

water into the groundwater layer (Wang et al., 2011; Zhang et al., 2018), thus making negative impacts on the annual streamflow (Liang et al., 2015; Yang et al., 2019; Wang et al., 2021). Therefore, we further explore the impact of considering continuous LAI dynamic in VIC model on the simulations of total evapotranspiration and soil moisture of top-most layer in the middle reaches with significant vegetation increase. The discrepancies between VIC simulations with



dynamic LAI and with fixed multi-year average LAI during 2000-2018 for annual total evapotranspiration and annual
average soil moisture were calculated respectively, as illustrated in the Figure 13. The model using dynamic LAI tends to
predict lower (higher) evapotranspiration and higher (lower) soil moisture than the model using static multi-year average
LAI in the year when LAI was lower (higher), and the discrepancies were especially large for maximum annual anomaly of
LAI, which is consistent with the findings of previous studies in the North America (Vivoni et al., 2008; Tang et al., 2012;
Liu et al, 2018). This could explain the less intense reduction in runoff when continuous LAI increase was not considered in
the hydrological simulation, as illustrated in the Figure 12.

Recent studies have increasingly focused on the effect of vegetation phenology and growth on runoff. It is found that earlier
spring phenology and delayed autumn phenology promote a longer growing season and can increase the period for plant
transpiration, potentially resulting in larger transpiration and might reduce the river runoff (Piao et al., 2019; Geng et al.,
2020; Wu et al., 2021; Chen et al., 2022). These results were consistent with the negative effect of intra-annual temporal
pattern of LAI associated with phenology change on runoff simulated by VIC model considering explicit vegetation
dynamics in this study.

## 5.4 Relationship between streamflow reduction and non-vegetation underlying surface change

To reduce sediment in the Yellow River, extensive water and soil conservation engineering measures including terraces and
check dams were constructed over the Loess Plateau for mitigating soil erosion and intercepting sediment. According to the
terrace proportion map (Cao et al., 2021) and statistical data about terrace areas of eight main tributaries (Liu et al., 2021),
built terrace was mainly distributed in the TNH-HYK (Figure 14(a)), and between 1979 and 2017 terrace areas of the Taohe,
Huangshui, Qingshuihe, Beiluohe, Fenhe, Jinhe, Weihe subbasins and TDG-LM increased by 4.5, 4.9, 2.6, 3.0, 20.8, 10.4,
4.2 and 1.4 times respectively (Figure 14(b)), which indicated that change intensities of terrace areas in the TNH-TDG and
LM-HYK were greater than that in the TDG-LM during the study period. Previous studies on the Loess Plateau have
confirmed that due to the slope land changes into flat land, terraces can damage the continuity of the slope and prolong the
infiltration time, resulting in a poor hydrological connectivity and obvious runoff reduction (Tian et al., 2003; Bai et al.,
2019). The study of Fu et al. (2020) also found that terrace plays critical role in reducing flood peak flow rate under extreme
rainstorms.

In addition, check dams were increasingly built for blocking sediment from hillslope into river channel, and the cumulative
number of dams built above Tongguan station during 1982-2015 was 3700 and 3010 for large-sized and medium-sized dams
respectively (Liu et al., 2020) (Figure 14(c)). Although check dam was originally designed to retain sediment, it still played
significant role in storing water for local crop irrigation, which has been captured by the significant increase of surface water
area derived from JRC product in the TNH-HYK, as shown in the Figure 14(d). It should be noted that greater change
intensities in the terrace area and surface water area in the TNH-TDG and LM-HYK comparing with that in the TDG-LM
could probably explain the greater impact of residual factors on the streamflow reduction (Figure 9) in the TNH-TDG and





LM-HYK, which is consistent with the spatial pattern of impacts of residual factors on the evapotranspiration increase for same sub-regions in the study of Wang et al. (2022).

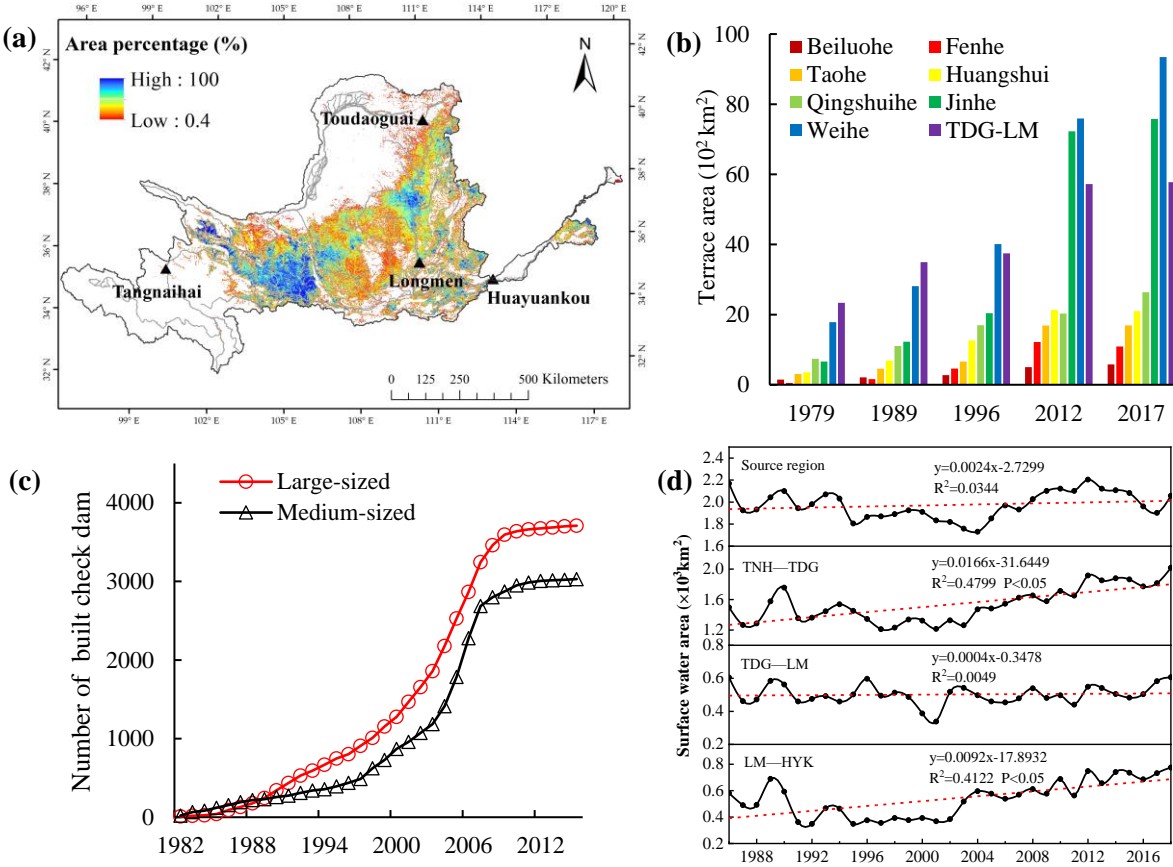

**Figure 14. (a) Spatial distribution of area percentage of the terrace at 1km resolution in 2018; (b) Terrace areas in different main watersheds from 1979 to 2017; (c) Total number of check dams built above Tongguan station from 1982 to 2015; (d) Total areas of permanent water bodies in source region, TNH-TDG, TDG-LM and LM-HYK during 1986-2019.**

For the source region where there are no significant changes in vegetation, terrace and check dam over last three decades, reported degradation of permafrost attributed to climate warming and human activities could enhance active layer thickness above permafrost and decrease duration of seasonally frozen ground (Wu and Zhang, 2008; Cheng and Jin, 2013). This would have profound effects on the hydrology by altering soil surface infiltration capacity and soil hydraulic conductivity (Jin et al., 2009, 2011). When permafrost is thawed, it can be changed from an aquitard to an aquifer in some areas and talik channels can be formed or enlarged, which facilitate surface water infiltration, river runoff decrease and groundwater recharge (Cheng and Jin, 2013). Cuo et al. (2013) have found it is highly possible that permafrost degradation has played a role in diminishing river runoff, meanwhile, increasing terrestrial water storage has also been confirmed in the study of Long et al. (2017).


## 5.5 Uncertainties

The gridded forcing data may introduce uncertainties in the simulations because these climate data are interpolated based on limited field observations. It would be better to merge high-accuracy microwave precipitation products and reanalysis data in the future. The GLASS LAI data were only used here, although differences exist between different LAI products, these LAI products generally consistent in the spatiotemporal changes across China (Piao et al., 2015; Zhu et al., 2016). Hence this would probably not change the general conclusions (Zhai et al, 2021).

The model parameters used in this study were calibrated using limited observations, and all grid cells of sub-region were characterized with constant parameter dataset based on an idealized assumption. Hence further calibration and validation should be conducted in more subbasins by collecting more naturalized hydrological data to mitigate this potential uncertainty which may influence the model performance. However, the calibration and validation results showed that VIC simulations matched observations well in various periods in each sub-regions (Figure 8), which ensures that the VIC model is applicable at regional scale in the YRB.

Scenario simulation method would unavoidably split the link and interaction between climate change and underlying surface change (Wu et al., 2017), which inevitably introduces a certain bias in quantifying the variation in streamflow induced by interannual and intra-annual changes of climate variables and vegetation. Even though the interactive effect was calculated by differencing the sum of variations in streamflow induced by climate and vegetation change and that induced by the coeffect in this study, this simplified method still cannot represent complicated feedback and response of climate and underlying surface change. Due to LAI increase is always associated with land cover change as a result of restoration projects, the vegetation's hydrological effect was considered as the total impact from LAI and land cover changes in this study. This inevitably involves the impacts of non-vegetated land cover conversion (e.g., urbanization), nevertheless this land cover change type only account for a very small proportion of YRB.

Due to the lack of water consumption data of coal mining, the effect of coal mining in analysing the relationship between non-vegetation underlying surface change and river runoff was not fully considered. In addition, only one model was applied here, and water and soil conservation engineering measures were not considered in the model. The analysis conclusions of this study should be proven in further studies by combing the statistical model, lumped model, distributed model and machine learning model.

## 6 Conclusion

YRB hydrological regimes have exhibited changes over the past decades as manifested by decreases in annual streamflow. Here, daily meteorological, monthly LAI and yearly land use/cover time-series data were coupled in the VIC hydrological model to disentangle the contributions from temporally explicit changes of climate variables and vegetation on the natural streamflow trend during 1982-2018. Comparing with the attribution of streamflow trend using the VIC simulation without considering dynamic LAI, simulations with dynamic LAI can better capture the temporally explicit variations of



evapotranspiration and soil moisture induced by vegetation, which enables VIC to reflect the cumulative effects of
vegetation changes on the streamflow. Results show that total effects from vegetation greening composed of interannual LAI
increase and intra-annual LAI temporal pattern change, primarily induced by large-scale ecological restoration, might play a
dominant role in the natural streamflow reduction of YRB over last decades. The impact from non-vegetation underlying
surface change is also great due to the water storage capacities of terraces and check dams. Positive contribution from
precipitation and wind speed almost offset the negative effect from temperature on the hydrological regimes. It should be
noted that the intra-annual precipitation temporal pattern change is able to affect the streamflow trend by altering the
precipitation intensity that is sensitive to the runoff in the YRB.

*Code and data availability*. VIC is open-source macroscale hydrological model (https://vic.readthedocs.io/en/master/).
Meteorological data was obtained from the China Meteorological Administration (http://data.cma.cn/). Time-series LAI data
was obtained from The Global Land Surface Satellite (GLASS) product (http://glass-product.bnu.edu.cn/). GLC_FCS30
product was downloaded from http://www.geodata.cn/. China soil map based harmonized world soil database (HWSD) (v1.1)
was download from http://data.tpdc.ac.cn/en/. The China terrace proportion map was download from
https://doi.org/10.5281/zenodo.3895585. Global surface water product was available from the Joint Research Centre (JRC)
(https://global-surface-water.appspot.com/download).

*Author contributions*. The paper has been authored by ZW with contributions from all the co-authors. ZW, QT and DW
contributed to the conceptualization and methodology. ZW ran the VIC model and performed the scenario simulations. PX
provided naturalized streamflow data and analysed its temporal dynamics. RX provided the meteorological dataset and a
high-performance computing platform to run VIC model. PX, RX, PS and FF contributed to the writing and revision of the
manuscript.

*Competing interests*. The contact author has declared that neither they nor their co-authors have any competing interests.

*Disclaimer*. Publisher's note: Copernicus Publications remains neutral with regard to jurisdictional claims in published maps
and institutional affiliations.

*Acknowledgements*. We are grateful to Beijing Normal University for providing long-term GLASS products. We thank Dr.
Zhang Xiao in the Aerospace Information Research Institute, Chinese Academy of Science for providing the land cover
products of different years. We also thank Dr. Zhang Xuejun in China Institute of Water Resources and Hydropower
Research for his assistance in the run and calibration of VIC model.



*Financial support.* This study was supported by the Joint Funds of the National Natural Science Foundation of China
(U2243210) and National Natural Science Foundation of China (41730645).

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
