# Peer review of "Attributing trend in naturalized streamflow to temporally explicit vegetation change and climate variation in the Yellow River Basin of China"

_Hydrology and Earth System Sciences, 2022_

## Author Comment (AC1)

Dear Editor,

On behalf of my co-authors, we thank you very much for giving us the opportunity to revise the manuscript (Manuscript ID: **hess-2022-196**). We appreciate the comments on our manuscript entitled "*Attributing trend in naturalized streamflow to temporally explicit vegetation change and climate variation in the Yellow River Basin of China*" by Zhihui Wang, Qiuhong Tang, Daoxi Wang, Peiqing Xiao, Runliang Xia, Pengcheng Sun, Feng Feng.

Great thanks to the reviewers and editors, we have revised the manuscript carefully according to the comments. All the changes were high-lighted in the revised manuscript and the point-by-point response to the comments of the reviewers is also listed below. Please let me know if you require any additional information on our paper.

Looking forward to hearing from you soon.

Best regards,

Zhihui Wang & Qiuhong Tang

Yellow River Institute of Hydraulic Research, Yellow River Conservancy Commission, China

Zhengzhou 450003, China

Email: wzh8588@aliyun.com

**Response to Comments from Reviewer 1**

**General comments:**

The manuscript assessed the effects from temporally explicit changes of climate variables and underlying surfaces on the streamflow trend using Variable Infiltration Capacity (VIC) model prescribed with continuously dynamic leaf area index (LAI) and land cover in Yellow River Basin. This study suggests that change in underlying surface has imposed a substantial trend on naturalized streamflow in Yellow River Basin. This topic is interesting and important for the water resources management in Yellow River Basin, especially for soil and water conservation measures and ecological restoration projects.

**Response:** Great thanks for your encouragement and recognition to our manuscript.

**Specific comments:**

**Point 1:** Line 156-159, the two-steps method was designed to consider time variant LAI in the VIC model simulation. Is it possible to use interannual change of LAI and land use in VIC model? In many hydrological models, the dynamical LAI and land use data are used. The version of the VIC model should be introduced.

**Response:** Great thanks for this comment. By default setting of VIC model, it only considers the climatology of vegetation (e.g., 12-month LAI), and the monthly LAI and land cover are stationary in different year during the simulation period, hence interannual change of LAI and land use cannot be considered in VIC model. The version of the VIC model used in this study is VIC 4.1.2.a. According to your comments, corresponding description has been revised, and the details are shown below.

By default setting of VIC model, it only considers the climatology of vegetation (e.g., 12-month LAI), and the monthly LAI and land cover are stationary in each year during the simulation period. Therefore, the impacts of continuous interannual change of LAI and land cover types on hydrological processes rarely be discussed in previous studies using VIC model (Xie et al., 2015; Yang et al., 2019; Zhai et al., 2021). In this study, the simulation scheme of VIC model (version 4.1.2.a) considering time-variant LAI was designed as the following two steps:

**Point 2:** Line 189, The monthly streamflow is evaluated by NSE, Bias and RMSE, which should be stated here, and calibration period and validation period, too.

**Response:** Great thanks for this comment. According to your suggestion, the NSE, Bias and RMSE have

been stated, and calibration period and validation period are also been clarified. Corresponding description and equations have been added, and the details are shown below.

To find the optimal parameter set, an optimization algorithm of the multi-objective complex evolution of the University of Arizona (MOCOM-UA) from Yapo et al. (1998) was implemented, and Nash–Sutcliffe efficiency (NSE), relative bias (Bias) and root mean square error (RMSE) were used as the objective function to assess the model performance, as illustrated in Eq.(1)- Eq.(3). The automatic calibration was carried out by running the VIC model thousands of times during calibration period (1980-1993), of which the first two years (1980–1981) used for warm up, and the period of 1994-1999 is the validation period.

$$NSE = 1 - \frac{\sum_{i=1}^{N}(Q_{obs,i} - Q_{sim,i})^2}{\sum_{i=1}^{N}(Q_{obs,i} - \overline{Q_{obs}})} \tag{1}$$

$$Bias = \frac{\sum_{i=1}^{N} Q_{sim,i} - \sum_{i=1}^{N} Q_{obs,i}}{\sum_{i=1}^{N} Q_{obs,i}} \tag{2}$$

$$RMSE = \sqrt{\frac{1}{N}\sum_{i=1}^{N}(Q_{sim,i} - Q_{obs,i})^2} \tag{3}$$

where $Q_{sim}$ and $Q_{obs}$ are the simulated and observed monthly streamflow, respectively, $\overline{Q_{obs}}$ is the arithmetic mean of the observed monthly runoff, i is the ith month, and N is the total number of months in calibration period.

**Point 3:** Line 281, the vegetation degradation in the source region and urbanization in the middle reaches. It's better to cite reference or data to support this attribution.

**Response:** Great thanks for this comment. Here is our mistake of the description. We just only want to descript the spatio-temporal change characteristics of interannual LAI during 1982-2018, without an aim to attribute the change trend of LAI. Decreasing LAI trend can be obviously seen in the source region in the Figure 4. Therefore, according to your great suggestion, corresponding description has been revised as below.

The downward LAI trend occurred in 15% of the basin which was mainly distributed in the source region.

**Point 4:** Line 315, the grey frames in Figure 7 are not necessary.

**Response:** Great thanks for this comment. It is better for displaying Figure 7 to remove the grey frames. According to your suggestion, corresponding figures have been revised, and the details are shown below.

[Figure]

**Figure 7. (a) The interannual change trend of annual runoff coefficiencts for 4 sub-regions, (b) precipitation-streamflow double mass curves for different sub-regions, and (c) precipitation-streamflow relationships in the two periods of 1982-1999 and 2000-2018 for 4 sub-regions.**

**Point 5:** Line 342, add Table 3.

**Response:** Great thanks for this comment. Here is our mistake of the description. We have revised this mistake, and the details are shown below.

As per performance criteria given by Moriasi et al. (2007), simulation results indicate that the VIC model has a good performance in simulating hydrological processes in not only subbasins and sub-regions.

**Point 6:** Line 335-346, a table that summarizes the value of NSE, RMSE and Bias at different gauges in different period is helpful.

**Response:** Great thanks for this comment. A summary of the value of NSE, RMSE and Bias at different drainage areas in different periods is helpful for readers to understand the performance of VIC model simulation. According to your suggestion, except for adding the Table 4, corresponding figure 8 and description have also been revised. The details are shown below.

The monthly hydrographs and average seasonal cycles of the simulated and naturalized streamflows for different catchment regions are shown in the Figure 8, and the accuracy metrics of all simulations in the Figure 8 are summarized in the Table 4.

[Figure]

**Figure 8. Comparisons of monthly streamflow and seasonal cycles of streamflow simulated by VIC and naturalized streamflow for different drainage areas during 1982-1999. (a) TNH, (b) TNH-TDG, (c) TDG-LM, (d) LM-HYK, (e) HYK**

**Table 4 Model performance metrics of monthly streamflows and seasonal cycles of streamflows in different drainage areas**

| Drainage areas | Monthly streamflow (1982-1999) | | | | | | Multi-year average of seasonal cycles of streamflow (1982-1999) | | |
| | Calibration period (1982-1993) | | | Validation period (1994-1999) | | | | | |
| | NSE | Bias | RMSE | NSE | Bias | RMSE | NSE | Bias | RMSE |
|---|---|---|---|---|---|---|---|---|---|
| TNH | 0.86 | 0.1% | 217.2 | 0.86 | 1.4% | 149.7 | 0.96 | -3.1% | 64.6 |
| TNH-TDG | 0.5 | 3.3% | 183.1 | 0.44 | 12.5% | 169.5 | 0.87 | -0.6% | 66.2 |
| TDG-LM | 0.63 | -11.7% | 77.7 | 0.63 | -7.0% | 82.2 | 0.67 | 1.9% | 48.5 |
| LM-HYK | 0.76 | -4.7% | 209.4 | 0.46 | -10.3% | 207.2 | 0.92 | 1.8% | 60.0 |
| HYK | 0.89 | -1.6% | 387.4 | 0.8 | 6.9% | 386.6 | 0.99 | -0.7% | 82.0 |

**Point 7:** Line 338, maybe the calibrated values of 6 soil parameters mentioned in Line 185 should be presented here.

**Response:** Great thanks for this comment. Here is a mistake of description in the number of calibrated soil parameters. We actually calibrated 7 parameters in the VIC model. These parameters have been added in the Table 3, and the details are shown below.

**Table 3. Calibrated parameters of VIC model for different drainage areas over YRB**

| Drainage areas | b | $D_s$ | $D_{smax}$ | $W_s$ | $b_1$ | $b_2$ | $b_3$ |
|---|---|---|---|---|---|---|---|
| TNH | 0.374 | 0.514 | 23.559 | 0.671 | 0.091 | 0.100 | 1.021 |
| TNH-TDG | 0.313 | 0.454 | 18.686 | 0.771 | 0.102 | 0.172 | 0.497 |
| TDG-LM | 0.135 | 0.056 | 7.427 | 0.354 | 0.264 | 0.824 | 1.107 |
| LM-HYK | 0.151 | 0.123 | 18.973 | 0.530 | 0.134 | 0.465 | 0.812 |

**Point 8:** Line 353, "For the HYK station, the contributions of all climate variables to the streamflow trend were positive excepting temperature, while larger negative effects from underlying surface change offset the slight positive effects of climate change on the streamflow trend (Figure 9).", it's better to move this sentence to Line 361 after the simulation result.

**Response:** Great thanks for this comment. According to your suggestion, this sentence has been move to the Line 361 after the simulation result, and the corresponding description has been revised as below.

From 1982 to 2018, the annual streamflow trend at HYK was $-3.71 \times 10^8$ m$^3 \cdot$yr$^{-1}$, of which changes in interannual precipitation (P_inter), temperature (T_inter), wind speed (WS_inter), intra-annual temporal pattern of precipitation (P_intra), interannual LAI (LAI_inter), intra-annual temporal pattern of LAI (LAI_intra), interactive effects of climate variables and vegetation (Interactive), and residual underlying surface (Resi.) accounted for 15.1% ($1.14 \times 10^8$ m$^3 \cdot$yr$^{-1}$), -23.5% ($-1.77 \times 10^8$ m$^3 \cdot$yr$^{-1}$), 8.7% ($0.66 \times 10^8$ m$^3 \cdot$yr$^{-1}$), 1.4% ($0.1 \times 10^8$ m$^3 \cdot$yr$^{-1}$), -26.6% ($-1.99 \times 10^8$ m$^3 \cdot$yr$^{-1}$) and -6% ($-0.45 \times 10^8$ m$^3 \cdot$yr-1), -3.5% ($-0.26 \times 10^8$ m$^3 \cdot$yr$^{-1}$), -15.2% ($-1.14 \times 10^8$ m$^3 \cdot$yr$^{-1}$), respectively. For the HYK station, the contributions of all climate variables to the streamflow trend were positive excepting temperature, while larger negative effects from underlying surface change offset the slight positive effects of climate change on the streamflow trend (Figure 9).

**Point 9:** Line 352-380, a table is needed to summarize the value of impacts and relative impacts rates shown in Figure 9.

**Response:** Great thanks for this comment. A summary of the value of impacts and relative impacts is very

helpful for readers to understand the Figure 9. Therefore, a new table 5 and corresponding description have been added, and the details are shown below.

The impacts and relative impact rates of eight influencing factors on the annual streamflow trends in different drainage areas were calculated using Eq.(6)-Eq.(14), as illustrated in the Figure 9 and Table 5.

**Table 5. Summary of the values of impacts and relative impacts rates of all influencing factors shown in the Figure 9**

| Influencing Factors | TNH | | TNH-TDG | | TDG-LM | | LM-HYK | | HYK | |
|---|---|---|---|---|---|---|---|---|---|---|
| | Impact ($10^8 m^3 \cdot yr^{-1}$) | Rate (%) | Impact ($10^8 m^3 \cdot yr^{-1}$) | Rate (%) | Impact ($10^8 m^3 \cdot yr^{-1}$) | Rate (%) | Impact ($10^8 m^3 \cdot yr^{-1}$) | Rate (%) | Impact ($10^8 m^3 \cdot yr^{-1}$) | Rate (%) |
| P_inter | 0.31 | 16.1% | 0.15 | 10.5% | 0.59 | 29.8% | 0.09 | 3.5% | 1.14 | 15.1% |
| T_inter | -0.72 | -38.0% | -0.48 | -33.2% | -0.24 | -12.1% | -0.33 | -12.3% | -1.77 | -23.5% |
| WS_inter | 0.22 | 11.5% | 0.18 | 12.8% | 0.13 | 6.5% | 0.13 | 4.7% | 0.66 | 8.7% |
| P_intra | 0.04 | 2.1% | -0.04 | -2.6% | -0.20 | -9.9% | 0.30 | 11.2% | 0.10 | 1.4% |
| LAI_inter | -0.16 | -8.3% | -0.23 | -16.3% | -0.60 | -30.5% | -1.00 | -37.5% | -1.99 | -26.6% |
| LAI_intra | -0.03 | -1.7% | -0.05 | -3.3% | -0.12 | -6.1% | -0.25 | -9.4% | -0.45 | -6.0% |
| Interactive | -0.02 | -0.8% | -0.02 | -1.6% | -0.06 | -3.1% | -0.16 | -6.0% | -0.26 | -3.5% |
| Resi. | -0.41 | -21.4% | -0.28 | -19.7% | -0.04 | -2.1% | -0.41 | -15.4% | -1.14 | -15.2% |

**Point 10:** Line 500, the slope land changes into the flat terraces could dramatically decrease the surface runoff generation and should not be ignored. It will also induce the change of intra-annual temporal pattern of LAI as shown in Figure 11(d).

**Response:** Great thanks for this comment. Yes, you are right. The slope land changes into the flat terraces could dramatically affect the vegetation growth by altering soil moisture, and thus interannual change and intra-annual temporal patten change of LAI could be induced by variations in micro-topography. According to your comment, this point has been added in the end of the Section 5.2, and details are shown below.

Due to phenology determines the start and end time of vegetation growth and is highly sensitive to climate change (Liang and Schwartz, 2009; Fu et al., 2019), climate warming has played an important role in advancing the spring phenology and delaying autumn phenology, and consequently extended the length of vegetation growing period across the globe (Piao et al., 2019; Menzel et al., 2020), especially for the semi-arid and semi-humid regions of China (Wu et al., 2015; Chen et al., 2022). In addition, the variations in micro-topography from slope land into

flat terrace significantly increase soil moisture (Bai et al., 2019), which could also inevitably alter inter-annual change and intra-annual temporal pattern of LAI.

**Point 11:** Line 519, Was the degradation of permafrost simulated in VIC model in this study? How about the setting? Please explain it.

**Response:** Great thanks for this comment. We didn't simulate the degradation of permafrost in VIC simulation in this study. Here we just want to discuss the possible underlying surface changes causing streamflow reduction (the residual factors in the Figure 9) in the source region according to previous studies. Some researchers have found it is highly possible that permafrost degradation has played a role in diminishing river runoff. According to your comment, corresponding discussion has been added in the Section 5.5 Uncertainties, and the details are shown below.

Due to the lack of water consumption data of coal mining and the effects of glaciers melting and permafrost degradation on the runoff generation were not considered during VIC simulation in this study, the impacts from coal mining, glacier and permafrost in analysing the relationship between non-vegetation underlying surface change and river runoff were not further clarified.

---

## Author Comment (AC2)

Dear Editor,

On behalf of my co-authors, we thank you very much for giving us the opportunity to revise the manuscript (Manuscript ID: **hess-2022-196**). We appreciate the comments on our manuscript entitled "*Attributing trend in naturalized streamflow to temporally explicit vegetation change and climate variation in the Yellow River Basin of China*" by Zhihui Wang, Qiuhong Tang, Daoxi Wang, Peiqing Xiao, Runliang Xia, Pengcheng Sun, Feng Feng.

Great thanks to the reviewers and editors, we have revised the manuscript carefully according to the comments. All the changes were high-lighted in the revised manuscript and the point-by-point response to the comments of the reviewers is also listed below. Please let me know if you require any additional information on our paper.

Looking forward to hearing from you soon.

Best regards,

Zhihui Wang & Qiuhong Tang

Yellow River Institute of Hydraulic Research, Yellow River Conservancy Commission, China

Zhengzhou 450003, China

Email: wzh8588@aliyun.com

**Response to Comments from Reviewer 2**

**General comments:**

Based on the Variable Infiltration Capacity (VIC) model prescribed with continuously dynamic leaf area index (LAI) and land cover, this study attributed the trend of naturalized streamflow to temporally explicit vegetation change and climate variation over the Yellow River Basin of China. They found that the effect of climate variation on streamflow is slight, while the change of underlying surface has imposed a substantial trend on naturalized streamflow. This research is of significance for understanding the underlying mechanisms of natural streamflow reduction, which can provide guidelines for local water resources management.

**Response:** Great thanks for your encouragement and recognition to our manuscript.

**Specific comments:**

**Point 1:** Equation 7: In scenario S3 ($f$ ($C_{inter}$, $P_{intra}$)), all climate variables and intra-annual temporal pattern of monthly precipitation vary according to observation records, while in scenario S2 ($f$ ($C_{inter}$)), only specific climate variable varied according to observation records, why the intra-annual temporal pattern of precipitation on the annual streamflow trend ($Q_{Pintra}$) can be calculated with equation 7? Maybe you should add a scenario (S2-1), in which all the interannual change of climate variables vary according to observation records while other variables vary according to control conditions in the S1. With this scenario, you can also check whether climate variables affect each other.

**Response:** Great thanks for this great comment. Here is an unclear description about original Equation 7 where $f$ ($C_{inter}$) represents simulated streamflow trend induced by interannual change of different climate variables, which make readers confused about scenario S2. Hence, three separated equations are shown in the revised description to illustrate how to calculate individual impact of precipitation, temperature and wind speed on the streamflow trend. In addition, we have checked the interactive effects of different climate variables on streamflow trend, and these effects are nearly close to the zero. Therefore, we adopted calculation formula that can make impacts of different four climate variables closed to the total impact of climate change. Hence according to your comments, corresponding description and equations have been revised, and the details are shown below.

To isolate the effect of climate variables on streamflow trend, we designed two scenarios. In Scenario S2, annual value of climate variable (precipitation, temperature and wind speed)

varied one by one according to observation records while other variables vary according to control conditions in the S1. In Scenario S3, annual values of all climate variables and intra-annual temporal pattern of monthly precipitation vary according to observation records while other variables vary according to control conditions in the S1. The impacts of climate variables were calculated as follows:

$$Q_{P_{inter}} = f(P_{inter}) - f(control) \tag{6}$$

$$Q_{T_{inter}} = f(T_{inter}, P_{inter}) - f(P_{inter}) \tag{7}$$

$$Q_{WS_{inter}} = f(WS_{inter}, T_{inter}, P_{inter}) - f(T_{inter}, P_{inter}) \tag{8}$$

$$Q_{P_{intra}} = f(WS_{inter}, T_{inter}, P_{inter}, P_{intra}) - f(WS_{inter}, T_{inter}, P_{inter}) \tag{9}$$

$$Q_C = Q_{P_{inter}} + Q_{T_{inter}} + Q_{WS_{inter}} + Q_{P_{intra}} = f(WS_{inter}, T_{inter}, P_{inter}, P_{intra}) - f(control) \tag{10}$$

Where $Q_{P_{inter}}$, $Q_{T_{inter}}$ and $Q_{WS_{inter}}$ are impacts of interannual change of precipitation, temperature and windspeed, respectively, and $Q_{P_{intra}}$ is impact of intra-annual temporal pattern of precipitation. $Q_C$ represents the total impacts of all climate variables. $f(control)$ and $f(P_{inter}, T_{inter}, WS_{inter}, P_{intra})$ are the simulated streamflow trends in the S1 and S3, and $f(P_{inter})$, $f(T_{inter}, P_{inter})$, and $f(WS_{inter}, T_{inter}, P_{inter})$ are the simulated streamflow trends in the S2.

2. It seems that the VIC simulations don't match well with the observations, and the Nash–Sutcliffe efficiency (NSE) of monthly streamflow is only 0.44 and 0.46 over TNH-TDG and LM-HYK during validation periods (Fig. 8). To ensure the accuracy of this research, it may be necessary to recalibrate the model parameters.

**Response:** Great thanks for this great comment. In the process of model calibration, we have used an optimization algorithm of MOCOM-UA, and the automatic calibration was carried out by running the VIC model 1000 times over calibration period (1980-1993), of which the first two years (1980–1981) used for warm up, and the period of 1994-1999 is the validation period. We have run the calibration program many times, and the optimal result was employed in this study. Therefore, we think these model parameters already are the best ones calibrated by the algorithm, and model parameters also have been shown in the Table 3. It can be seen that from Table 4, NSE in the TNH-TDG and LM-HYK in the calibrate period is very high, merely NSE is slightly lower than the value of 0.5 in the validation period,

which might be acceptable for model simulation.

There are some reasons for this. Actually, it is very difficult to accurately acquire naturalized streamflow due to high uncertainties of human water use data, especially from irrigation, which could explain the NSE slightly lower than 0.5 in the validation period in the TNH-TDG and LM-HYK where there are large irrigated areas. In the future, the high-quality naturalized data and hydrological simulation considering irrigation should be used to mitigate uncertainties of model parameters. The corresponding descriptions have been added in the section Uncertainties as below.

**Table 3.   Calibrated parameters of VIC model for different drainage areas over YRB**

| Drainage areas | b | $D_s$ | $D_{smax}$ | $W_s$ | $b_1$ | $b_2$ | $b_3$ |
|---|---|---|---|---|---|---|---|
| TNH | 0.374 | 0.514 | 23.559 | 0.671 | 0.091 | 0.100 | 1.021 |
| TNH-TDG | 0.313 | 0.454 | 18.686 | 0.771 | 0.102 | 0.172 | 0.497 |
| TDG-LM | 0.135 | 0.056 | 7.427 | 0.354 | 0.264 | 0.824 | 1.107 |
| LM-HYK | 0.151 | 0.123 | 18.973 | 0.530 | 0.134 | 0.465 | 0.812 |

It is difficult to accurately acquire naturalized streamflow due to some uncertainties of human water use data, especially from irrigation, which could explain the NSE lower than 0.5 in the validation period (Table 4) in the TNH-TDG and LM-HYK where there are large irrigated areas. In addition, all grid cells of sub-region were characterized with constant parameter dataset based on an idealized assumption. Hence further calibration should be conducted in more subbasins by collecting high-quality naturalized hydrological data and using hydrological model considering irrigation to mitigate uncertainties of model parameters.

3. Why only the results over TDG-LM and LM-HYK are shown in Fig. 12, and what's the streamflow trend over other regions?

**Response:** Great thanks for this great comment. In this Section, we just want to discuss the discrepancy of simulated annual streamflow trend based on VIC considering and without considering continuous LAI dynamics to demonstrate the implication of considering temporally explicit vegetation change on runoff simulation using VIC. Therefore, in order to demonstrate this discrepancy clearly and obviously we just only selected two typical sub-regions (TDG-LM and LM-HYK) in the Middle Reaches, where large-scale and intensive ecological restoration has been implemented since 1999 instead of Upper Reaches with slight vegetation change.

To explore the discrepancy in evaluating the hydrological effect of vegetation using VIC considering and without considering temporally explicit LAI change, we calculated the annual streamflow trend

change by differencing simulation of scenario S1 and simulation with dynamic annual LAI observations while other variables varied under control conditions in the S1, and then calculated the streamflow trend change using the combination of scenario S1 and simulation where annual LAI during 1982-1999 and 2000-2018 were fixed into the multi-year averages of corresponding periods respectively, while other variables varied same as S1. Likewise, the annual streamflow trend changes simulated by continuous and noncontinuous change of intra-annual temporal pattern of LAI were also calculated using same way.

It should be emphasized for peer review expert here that Figure 12 illustrates the simulated annual streamflow using VIC considering and without considering continuous dynamics of interannual LAI and intra-annual temporal pattern of LAI in the TDG-LM (Figure 12(a)~(b)) and LM-HYK (Figure 12(c)~(d)) instead of the original naturized annual streamflow trend, which can be seen in the Figure 3. Corresponding figure has been revised as below.

[Figure]

**Figure 12. The comparison of simulated annual streamflow trend using VIC considering and without considering continuous dynamics of interannual LAI (a and c) and intra-annual temporal pattern of LAI (b and d) in the TDG-LM and LM-HYK. The insets show the time-series of difference between simulated annual streamflow with VIC considering and without considering continuous LAI dynamics, and its significance level of change trend.**

4. To reduce the uncertainty of this research, it's better to show the multi-years average evapotranspiration and soil moisture in Fig. 13, rather than a specific year. In addition, please explain the meaning of these line charts in the manuscript.

**Response:** Great thanks for this great comment. Let me explain for you. Figure 13 is not annual ET and soil moisture simulated by VIC, but the difference between VIC simulations with dynamic LAI and with fixed multi-year average LAI during 2000-2018. This difference was calculated in order to demonstrate the implication of considering temporally explicit vegetation change on ET and soil moisture simulation using VIC. In the Figure 13, the reason why we selected three specific years of 2000 with low LAI, 2010 with medium LAI, and 2018 with high LAI is that we just want to clearly show the discrepancy between ET and soil moisture simulated by VIC considering LAI dynamics and without considering LAI dynamics. According to the discrepancy in different year with different level of LAI shown in the Figure 13, the model using dynamic LAI tends to predict lower (higher) evapotranspiration and higher (lower) soil moisture than the model using static multi-year average LAI in the year when LAI was lower (higher). This could explain the less intense reduction in runoff when continuous LAI increase was not considered in the hydrological simulation. Corresponding figure has been revised as below.

[Figure]

**Figure 13. The difference between two simulations by VIC with dynamic LAI and fixed multi-year average LAI during 2000-2018 for annual total evapotranspiration (a) and annual average soil moisture (b) in the**

**middle reaches in the year of 2000, 2010 and 2018. The insets show the statistical histogram of the difference value.**

**Minor Comments**

5. Line 29: "the effect climate variation on streamflow" should be changed to "the effect of climate variation on streamflow".

**Response:** Great thanks for this great comment. According to your suggestion, corresponding description has been revised, and the details are shown below.

Overall, the effect of climate variation on streamflow is slight because positive effect from precipitation and wind speed changes was offset by the negative effect from increasing temperature.

6. Line 129: "hman" should be changed to "human".

**Response:** Great thanks for this great comment. According to your suggestion, corresponding description has been revised, and the details are shown below.

Naturalized runoff at the target gauge was estimated by adding human water use data from irrigation, industrial and domestic sectors over the drainage area of the target gauge back to the observed runoff at the target gauge (Yuan et al., 2017; Zhang et al., 2020)

7. Line 267: "HKY" should be changed to "HYK".

**Response:** Great thanks for this great comment. According to your suggestion, corresponding description has been revised, and the details are shown below.

Temporally, all monthly streamflow experienced negative trends at HYK station, with a greatest reduction (18.6%) was found in August.

8. Fig. 4a: The label of the colorbar is incorrect. "-6" should be changed to "6".

**Response:** Great thanks for this great comment. According to your suggestion, corresponding figure has been revised, and the details are shown below.

[Figure]

9. Are the trends in Fig. 7, 10, and 12 significant? It's better to add the confidence interval in the figures.

**Response:** Great thanks for this great comment. The trends in the Figure 7 are all significant. The △P25 annual time-series at most meteorological stations in the Figure 10 show significant change trend. It should be noted that although the change trends of simulated annual streamflow (Figure 12(b)~(d)) are insignificant due to original interannual fluctuations were reserved in the scenario simulations, these trends become more significant when continuous LAI dynamics were considered in VIC simulation, and time-series of difference between simulated annual streamflow with VIC considering and without considering LAI dynamics show extremely significant change trend (P<0.001). To make the figures more clear, understandable and scientific, the significance level of the trends in the Figure 7, 10 and 12 were added in the original figures, and the details are shown below.

[Figure]

[Figure]

**Figure 7. (a) The interannual change trend of annual runoff coefficiencts for 4 sub-regions, (b) precipitation-streamflow double mass curves for different sub-regions, and (c) precipitation-streamflow relationships in the two periods of 1982-1999 and 2000-2018 for 4 sub-regions.**

[Figure]

**Figure 10. The impacts ($\triangle P_{25}$) of changes in interannual precipitation (a) and intra-annual monthly to annual precipitation ratio (b) on the P25 trend of each station. Hollow stars show $\triangle P25$ time-series with statistically significant trends (P<0.01)**

[Figure]

[Figure]

**Figure 12. The comparison of simulated annual streamflow trend using VIC considering and without considering continuous dynamics of interannual LAI (a and c) and intra-annual temporal pattern of LAI (b and d) in the TDG-LM and LM-HYK. The insets show the time-series of difference between simulated annual streamflow with VIC considering and without considering continuous LAI dynamics, and its significance level of change trend.**

10. Fig. 12: "LM-TDG" should be changed to "LM-HYK" in the figure caption.

**Response:** Great thanks for this great comment. According to your suggestion, corresponding caption has been revised, and the details are shown below.

Figure 12. The comparison of simulated annual streamflow trend using VIC considering and without considering continuous dynamics of interannual LAI (a and c) and intra-annual temporal pattern of LAI (b and d) in the TDG-LM and LM-HYK. The insets show the time-series of difference between simulated annual streamflow with VIC considering and without considering continuous LAI dynamics, and its significance level of change trend.

---

## Author Comment (AC3)

Dear Editor,

On behalf of my co-authors, we thank you very much for giving us the opportunity to revise the manuscript (Manuscript ID: **hess-2022-196**). We appreciate the comments on our manuscript entitled "*Attributing trend in naturalized streamflow to temporally explicit vegetation change and climate variation in the Yellow River Basin of China*" by Zhihui Wang, Qiuhong Tang, Daoxi Wang, Peiqing Xiao, Runliang Xia, Pengcheng Sun, Feng Feng.

Great thanks to the reviewers and editors, we have revised the manuscript carefully according to the comments. All the changes were high-lighted in the revised manuscript and the point-by-point response to the comments of the reviewers is also listed below. Please let me know if you require any additional information on our paper.

Looking forward to hearing from you soon.

Best regards,

Zhihui Wang & Qiuhong Tang

Yellow River Institute of Hydraulic Research, Yellow River Conservancy Commission, China

Zhengzhou 450003, China

Email: wzh8588@aliyun.com

**Response to Comments from Reviewer 3**

**General comments:**

Attributing hydrological variation to climate and surface feature is a hot topic under global warming in Anthropocene and the Yellow River basin is a typical case where eco-hydrological processes have changed severely during recent decades, so it's critical to quantify the contributions of land use/cover and climate changes to streamflow reduction in the YRB, China. Many previous studies have already focused on this topic using various methods and have got some interesting findings. But as authors of this MS mentioned, those studies analyzed the contributions of inter-annual change of driving factors to hydrological processes, while effects of the intra-annual change of climate and vegetation have not been examined. To solve it, this MS improved the VIC model by coupling time-series land cover and LAI remote sensing data to capture the cumulative effect of dynamic vegetation on the hydrological cycle, and designed six scenario simulation experiments to separate impacts of intra-annual changes of climate and vegetation from those of inter-annual changes and the interactive effects. Since topic of this MS is meaningful and innovative, methods are convincing, and results & discussions are reasonable, I recommend it to be published with revisions. Some specific comments are as follows:

**Response:** Great thanks for your encouragement and recognition to our manuscript.

**Specific comments:**

**Point 1:** Line 187-188, the optimization algorithm named MOCOM-UA is inconsistent with SCE-UA in fig. 2.

**Response:** Great thanks for this great comment. Here is an obvious mistake when we draw this flowchart. Corresponding figure has been revised, and the details are show below.

[Figure]

Figure 2. The flowchart of VIC model setup considering temporally explicit vegetation change

**Point 2:** Line 313, unit of runoff coefficient is wrong in Fig. 7a.

**Response:** Great thanks for this great comment. Here is a mistake when we draw this figure. We have revised according to your suggestion, and the details are shown below.

[Figure]

**Point 3:** Line 365, as Fig. 9 shows, temperature has negative impact on streamflow in the source regions. This is inconsistent with my understanding of this region that higher temperature contributes to an increase in runoff due to its role in promoting glacier melting, although the authors discussed the negative impact in Section 5.4 and attributed it to permafrost degradation.

**Response:** Great thanks for this great comment. In this study, glacier melting and permafrost degradation were not considered during VIC simulation, hence the negative impact of temperature on streamflow in the Figure 9 represents its direct impact caused by ET increase driven by temperature increase, and the indirect impact of temperature increase by altering glacier and permafrost condition on the annual runoff are not further discussed in this study because of the topic of this paper is to figure out the impacts of temporally explicit vegetation changes on runoff reduction. According to previous studies, glacier and snow melting indeed contribute to an increase in runoff in the Qinghai-Tibet plateau region, whereas the area ratio of glaciers in the source region of YRB is relatively low, and the negative effect from permafrost degradation probably offset the slight positive effect from glaciers melting, which is a very interesting

scientific problem worthy of study in the future. Therefore, we added some discussion about this aspect in the section of "Uncertainties",

Due to the lack of water consumption data of coal mining and the effects of glaciers melting and permafrost degradation on the runoff generation were not considered during VIC simulation in this study, the impacts from coal mining, glacier and permafrost in analysing the relationship between non-vegetation underlying surface change and river runoff were not further clarified.

**Point 4:** Line 390 It's confusing to use rainfall and precipitation concurrently because they have different meanings and precipitation includes rainfall, snow, hail, etc.

**Response:** Great thanks for this great comment. We have revised all descriptions of "rainfall" into "precipitation" according to your suggestion, and the details are shown below.

Due to runoff yield in excess of infiltration is the dominant runoff mechanism where precipitation intensity is the crucial driving force over the most of YRB (Jin et al., 2020), we then focused on the impacts of interannual precipitation and intra-annual monthly to annual precipitation ratio on the precipitation intensity.

Precipitation frequency caused by temporal pattern change of precipitation possibly influence the hydrological process over the YRB.

**Point 5:** Line 435-439 There are some grammatical errors in these sentences such as vegetation phenology rather than phenological.

**Response:** Great thanks for this great comment. According to your suggestion, corresponding description has been revised, and the details are shown below.

Therefore, the massive vegetation type conversion from cropland into forest-grass vegetation could significantly alter the vegetation phenology, which could lead to the interannual trend of intra-annual monthly to annual LAI ratio increased in the spring and decreased in the summer (Figure 6).

**Point 6:** Line 455 Error in name of y-axis, annual instead of anuual.

**Response:** Great thanks for this great comment. According to your suggestion, corresponding figure has been revised, and the details are shown below.

[Figure]

**Figure 12. The comparison of simulated annual streamflow trend using VIC considering and without considering continuous dynamics of interannual LAI (a and c) and intra-annual temporal pattern of LAI (b and d) in the TDG-LM and LM-HYK. The insets show the time-series of difference between simulated annual streamflow with VIC considering and without considering continuous LAI dynamics, and its significance level of change trend.**

**Point 7:** Line 455 Differences between trends of annual streamflow with continuous and noncontinuous LAI changes is little visually, especially for Figs. 12b, c, and d, so adding significant level of these trends may be more convincing.

**Response:** Great thanks for this comment. Due to original interannual fluctuations of annual streamflow were reserved in the scenario simulations, the change trends of simulated annual streamflow (Figure 12(b)(c)(d)) are insignificant. It is also shown from the Figure 12 that the range of fluctuation of original annual streamflow is relatively large, hence differences between trends of annual streamflow with continuous and noncontinuous LAI changes is little visually. However, it should be noted that time-series of difference between simulated annual streamflow with VIC considering and without considering LAI dynamics actually shows extremely significant change trend (P<0.001), and this insert plots has been added into the Figure 12 to demonstrate obvious differences. To make the figures more clear,

understandable and scientific, the significance level of the trend of time-series of difference in the Figure 12 was added in the original figures, and the details are shown below.

[Figure]

Figure 12. The comparison of simulated annual streamflow trend using VIC considering and without considering continuous dynamics of interannual LAI (a and c) and intra-annual temporal pattern of LAI (b and d) in the TDG-LM and LM-HYK. The insets show the time-series of difference between simulated annual streamflow with VIC considering and without considering continuous LAI dynamics, and its significance level of change trend.

**Point 8:** Line 458 Error in figure name; Figs. 12c and 12d are for LM-HYK instead of LM-TDG.

**Response:** Great thanks for this great comment. According to your suggestion, corresponding description has been revised, and the details are shown below.

Figure 12. The comparison of simulated annual streamflow trend using VIC considering and without considering continuous dynamics of interannual LAI (a and c) and intra-annual temporal pattern of LAI (b and d) in the TDG-LM and LM-HYK. The insets show the time-series of difference between simulated annual streamflow with VIC considering and without

considering continuous LAI dynamics, and its significance level of change trend.

**Point 9:** Line 486-491 This paragraph should move to Section 5.2 to build a direct connection with streamflow change, so that take Section 5.3 as an additional methodological analysis to highlight the role of vegetation dynamics in streamflow trend.

**Response:** Great thanks for this great comment. According to your suggestion, corresponding paragraph has been moved to the end of the Section 5.2, and the details are shown below.

Recent studies have increasingly focused on the effect of vegetation phenology and growth on runoff. It is found that earlier spring phenology and delayed autumn phenology promote a longer growing season and can increase the period for plant transpiration, potentially resulting in larger transpiration and might reduce the river runoff (Piao et al., 2019; Geng et al., 2020; Wu et al., 2021; Chen et al., 2022). These results were consistent with the negative effect of intra-annual temporal pattern of LAI associated with phenology change on runoff simulated by VIC model considering explicit vegetation dynamics in this study.

**Point 10:** Line 544 Usage of due to is wrong, please check it throughout this MS.

**Response:** Great thanks for this great comment. According to your suggestion, corresponding description has been revised, and the details are shown below.

The LAI increase is always associated with land cover change as a result of restoration projects, hence the vegetation's hydrological effect was considered as the total impact from LAI and land cover changes in this study.

**Point 11:** Line 547 Grammatical error, accounts instead of account.

**Response:** Great thanks for this great comment. According to your suggestion, corresponding description has been revised, and the details are shown below.

This inevitably involves the impacts of non-vegetated land cover conversion (e.g., urbanization), nevertheless this land cover change type only accounts for a very small proportion of YRB.

**Point 12:** Line 555-557 Usage of disentangle is wrong, rewritten it, please.

**Response:** Great thanks for this great comment. According to your suggestion, corresponding description has been revised, and the details are shown below.

Here, daily meteorological, monthly LAI and yearly land use/cover time-series data were coupled in the VIC hydrological model to clarify the contributions from temporally explicit changes of climate variables and vegetation on the natural streamflow trend during 1982-2018.